# Glutaredoxin 1 controls monocyte reprogramming during nutrient stress and protects mice against obesity and atherosclerosis in a sex-specific manner

Yong Joo Ahn[1,5], Luxi Wang [1,5], Sina Tavakoli [2], Huynh Nga Nguyen[3], John D. Short[4] & Reto Asmis [1,3✉]

High-calorie diet-induced nutrient stress promotes thiol oxidative stress and the reprogramming of blood monocytes, giving rise to dysregulated, obesogenic, proatherogenic monocyte-derived macrophages. We report that in chow-fed, reproductively senescent female mice but not in age-matched male mice, deficiency in the thiol transferase glutaredoxin 1 (Grx1) promotes dysregulated macrophage phenotypes as well as rapid weight gain and atherogenesis. Grx1 deficiency derepresses distinct expression patterns of reactive oxygen species and reactive nitrogen species generators in male versus female macrophages, poising female but not male macrophages for increased peroxynitrate production. Hematopoietic Grx1 deficiency recapitulates this sexual dimorphism in high-calorie diet-fed LDLR$^{-/-}$ mice, whereas macrophage-restricted overexpression of Grx1 eliminates the sex differences unmasked by high-calorie diet-feeding and protects both males and females against atherogenesis. We conclude that loss of monocytic Grx1 activity disrupts the immunometabolic balance in mice and derepresses sexually dimorphic oxidative stress responses in macrophages. This mechanism may contribute to the sex differences reported in cardiovascular disease and obesity in humans.

[1] Department of Internal Medicine, Wake Forest School of Medicine, Winston-Salem, NC, USA. [2] Departments of Radiology and Medicine, University of Pittsburgh, Pittsburgh, PA, USA. [3] Department of Biochemistry, University of Texas Health Science Center at San Antonio, San Antonio, TX, USA. [4] Department of Pharmacology, University of Texas Health Science Center at San Antonio, San Antonio, TX, USA. [5] These authors contributed equally: Yong Joo Ahn, Luxi Wang. ✉email: rasmis@wakehealth.edu

Glutaredoxin 1 (Grx1) belongs to a family of thiol trans-ferases, which specifically catalyze the reduction of mixed disulfides between glutathione (GSH) and protein thiols[1,2]. The formation of these mixed disulfides, referred to as "protein *S*-glutathionylation", is a reversible post-translational modification of cysteine residues, and represents a major redox signaling mechanisms that play a critical role in regulating monocyte and macrophage functions[1–5]. Protein *S*-glutathiony-lation in most instances leads to loss of function of the protein, and in some cases, its degradation[1–4]. We showed that chronic metabolic stress in mice alters the *S*-glutathionylation status of over 120 proteins in macrophages and promotes the repro-gramming of their proteome[6], giving rise to dysfunctional, proatherogenic macrophages. Furthermore, atherosclerotic lesions in mice show a dramatic increase in *S*-glutathionylated proteins, particularly within macrophages[7]. However, the underlying mechanisms involved in metabolic stress-induced *S*-glutathionylation[8] and proteome reprogramming remained unknown. We reported that overexpressing Grx1 protects mac-rophages against nutrient stress-induced protein *S*-glutathiony-lation and completely prevents "priming" and dysfunction in vitro[5,6]. The critical roles that monocytes and macrophages play in the development of metabolic diseases, including obesity, insulin resistance, and atherosclerosis, are well established[9–12]. Furthermore, gene-disease association data suggest a strong association of *Glrx1* with atherosclerosis and heart disease[13], but the roles of Grx1 in monocyte and macrophage biology and chronic inflammatory diseases are not known.

Here we used Grx1-deficient mice, bone marrow transplan-tation and gene transfer approaches as well as gene profiling to elucidate the roles of Grx1 in monocytes and macrophages in the context of aging and diet-induced atherogenesis, weight gain and adipose tissue inflammation. Our data identified a critical protective role for monocytic Grx1 in atherosclerosis and obe-sity and revealed a negative feedback mechanism between Grx1 activity and the expression of NADPH oxidases (NOX), nitric oxide synthases (NOS), and arginase 2 (ARG2). Our findings suggest that in addition to maintaining the immunometabolic balance by regulating the thiol redox status of the macrophage proteome[6,14], Grx1 also controls the production of reactive oxygen species (ROS) and reactive nitrogen species (RNS) in macrophages in a sexually dimorphic fashion. This mechanism may contribute to the increased obesity and cardiovascular risks observed in postmenopausal women.

## Results and discussion

**Metabolic phenotype of Grx1 deficiency.** To evaluate the effects of Grx1 deficiency on the metabolic phenotype of aging mice, WT and Grx1$^{-/-}$ mice were fed chow diet for 18 months and their body weights and fasted blood glucose levels were monitored weekly. While male WT and Grx1$^{-/-}$ mice showed very similar weight gains over the 18-month period (Fig. 1a), female Grx1$^{-/-}$ at 6 months of age, showed a significant acceleration in weight gain up to 14 months of age compared to WT mice, at which point their body weights plateaued. At one year of age, female Grx1$^{-/-}$ were grossly obese and 30% heavier than the corresponding male mice. Blood glucose levels were identical in male WT and Grx1$^{-/-}$ mice throughout the 18 months. In contrast, female Grx1$^{-/-}$ mice showed elevated blood glucose levels at around 8 months of age, two months after the onset of accelerated weight gain, indicating they began to develop insulin resistance in response to the increased weight gain (Fig. 1c, d). Analysis of organ weights suggests that the weight gain occurred primarily in the adipose tissue; the gonadal fad pads, being the largest fat pad in mice[15], showed a 4.2-fold increase in weight (Supplemental Fig. 1a).

While we did not observe any differences in the weights of either hearts or kidneys of female WT and Grx1$^{-/-}$ mice, liver weights were 21% higher in female Grx1$^{-/-}$ mice (Supplemental Fig. 1b, c and d), indicating the possible onset of liver steatosis. As expec-ted, organ weights were identical between male WT and Grx1$^{-/-}$ mice (Supplemental Fig. 1a–d). This was surprising, as Shao et al. had reported that male Grx1$^{-/-}$ mice on a C57BL/6NJ genetic background fed a normal chow diet not only became obese at 8 months of age but also developed hyperglycemia and fatty liver disease[16]. The explanation for these dramatic differences in metabolic phenotype between Shao's male C57BL/6NJ Grx1$^{-/-}$ mice and ours, which were backcrossed into the C57BL/6 J genetic background, are likely to be found in the genomic var-iations between these two strains. Our C57BL6/J mice carry an in-frame five-exon deletion within the nicotinamide nucleotide transhydrogenase (NNT) gene, which accounts for their glucose intolerance[17]. In addition to being NNT competent mice, C57BL6/NJ mice also show other genomic differences and differ from the C57BL6/J strain in a number of phenotypic traits[18], which may account for the different metabolic phenotypes reported in the literature. However, we are the first to report that Grx1 deficiency unmasks a dramatic sexual dimorphism with regard to the metabolic phenotype of aging C57BL6/J mice. Whether this sexual dimorphism is lost in aging C57BL6/NJ Grx1$^{-/-}$ mice is not known, as Shao et al. only reported on male C57BL6/NJ Grx1$^{-/-}$ mice[16].

To investigate the potential mechanisms underlying the accelerated weight gain in female Grx1$^{-/-}$ mice, we assessed plasma lipid and lipoprotein compositions in these aged mice. In 18-month-old mice, we found no significant differences in total plasma triglyceride levels between aged WT and Grx1$^{-/-}$ mice (not shown). Compared to WT mice, total cholesterol was modestly elevated, by 6.7% in male Grx1$^{-/-}$ mice and by 17.2% in Grx1$^{-/-}$ female mice (Fig. 1e). Grx1 deficiency had no effect on lipoprotein profiles in aged male mice (Supplemental Fig. 2). Interestingly, the lipoprotein profiles revealed that the elevated cholesterol levels in female Grx1$^{-/-}$ mice were restricted to the triglyceride-rich VLDL fractions (Fig. 1f). However, these minor changes in total plasma cholesterol alone cannot explain the dramatic effect of Grx1 deficiency on the metabolic phenotype of female Grx1$^{-/-}$ mice. Grx1 deficiency, in either male or female mice, also did not significantly alter WBC, RBC or platelet counts (Supplemental Table 1) nor blood monocyte, lymphocyte or neutrophil counts (Supplemental Fig. 3a, b), ruling out effects of Grx1 deficiency on hematopoiesis.

**Adipose tissue inflammation and atherogenesis in aged Grx1-deficient female mice is driven by monocyte priming and enhanced recruitment of monocyte-derived macrophages.** We recently demonstrated in baboons that even moderate high-calorie diet-induced elevations in blood cholesterol promote monocyte "priming" and dysfunction[19], i.e. the conversion of healthy blood monocytes into a hyper-chemotactic phenotype, which is reprogrammed and poised to differentiate into proin-flammatory and proatherogenic macrophages[7,20]. To examine whether monocytes were primed in aged Grx1$^{-/-}$ mice and to assess the responsiveness of blood monocytes to chemoat-tractants, we implanted Matrigel plugs loaded with the monocyte chemoattractant MCP-1 into the flanks of each mouse[3,7,8]. After three days, the plugs were removed, enzymatically dissolved and macrophages were counted. While we saw no statistical difference in the number of monocyte-derived macrophages in plugs from aged male WT and Grx1$^{-/-}$ mice, plugs from female Grx1$^{-/-}$ mice showed a 2.3-fold higher macrophage content than those removed from female WT mice (Fig. 2a), suggesting that

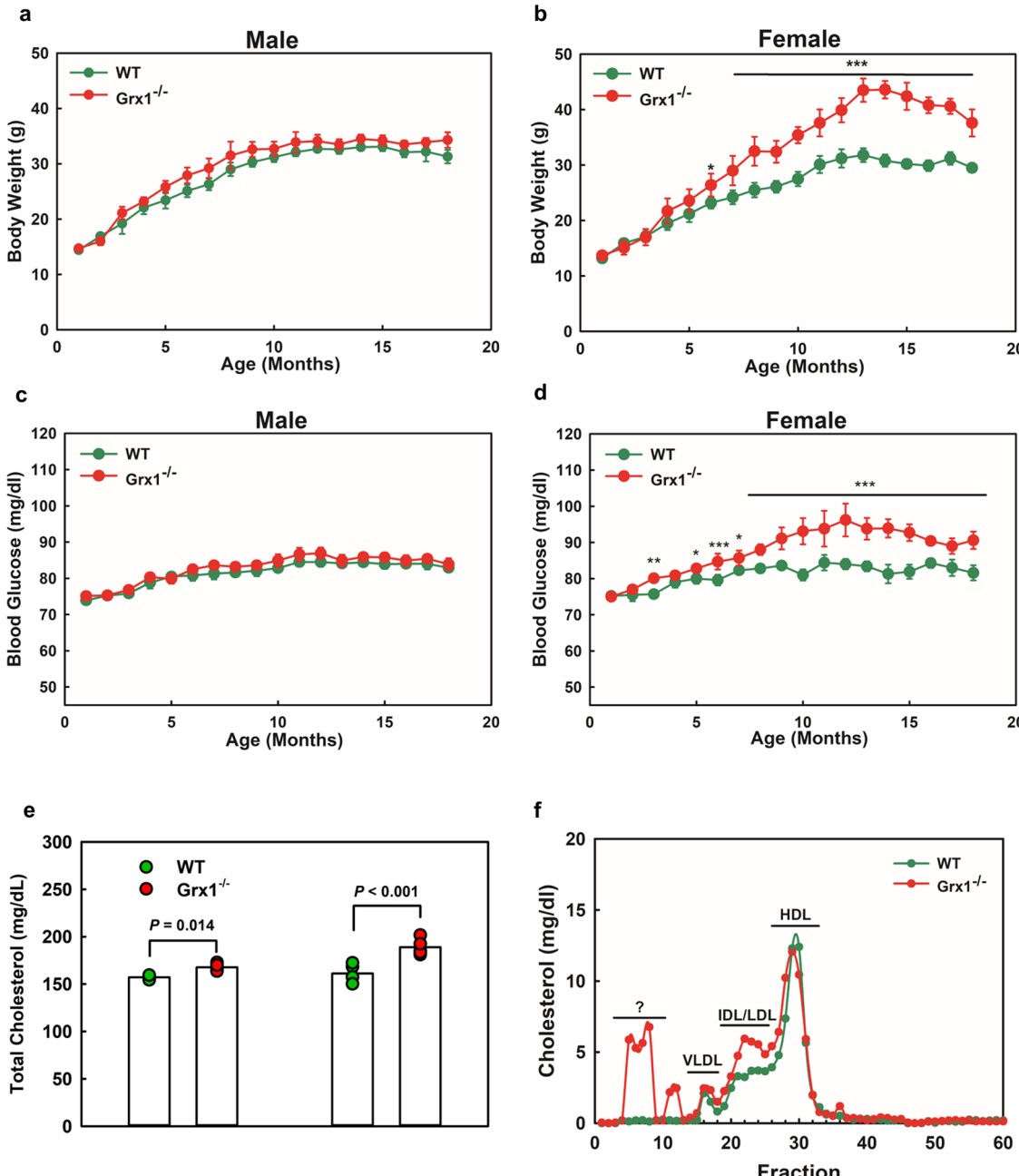

**Fig. 1 Characterization of metabolic phenotypes of chow-fed aged male and female Grx1$^{-/-}$ mice as compared to age-matched male and female C57BL/6 J mice. a–b** Body weights. **c–d** Fasting blood glucose levels. **e–f** Total plasma cholesterol levels and plasma lipoprotein profiles in female WT ($n = 5$) and Grx1$^{-/-}$ mice ($n = 4$) at 18 months of age. Lipoprotein profiles were generated from pooled plasma samples. All data are expressed as mean ± S.D., $n = 5–6$ mice per group for **a–f**; *One-way ANOVA followed by Fisher's Least Significance Difference test was used to compare the mean values between experimental groups. $P < 0.05$; **$P < 0.01$; ***$P < 0.001$; Source data are provided as a Source Data file.

monocytes from aged female Grx1$^{-/-}$ mice are indeed primed and hyper-chemotactic. To examine whether increased monocyte priming and hyper-chemotactic activity induced by Grx1-deficiency in female mice (Fig. 2a) was sufficient to promote the recruitment of monocyte-derived macrophages into their adipose tissue (AT), we quantified F4/80 mRNA levels in gonadal fat pads isolated from these aged mice. Indeed, F4/80 mRNA levels were increased 9.2-fold in gonadal fat pads from aged females Grx1$^{-/-}$ mice (Fig. 2b), suggesting massive macrophage infiltration and thus the onset of AT inflammation. Furthermore, H&E staining of paraffin section of these fat pads identified large crown-like structures in white AT (WAT) from aged female

Grx1$^{-/-}$ mice (Fig. 2d), hallmarks of leukocyte and macrophage infiltration and adipose inflammation[21]. These structures were not seen in aged female WT mice (Fig. 2c) or in aged male WT of Grx1$^{-/-}$ mice (not shown). These latter three groups of mice also showed no difference in F4/80 mRNA levels (Fig. 2b), indicating that the macrophage content in the WAT is identical in aged male WT of Grx1$^{-/-}$ and female WT mice. These findings are in good agreement with our observation that monocytes from these three groups of mice also showed no significant increase in chemotactic activity (Fig. 2a).

Since the recruitment of monocyte-derived macrophages is essential and rate-limiting for atherogenesis, we next examined

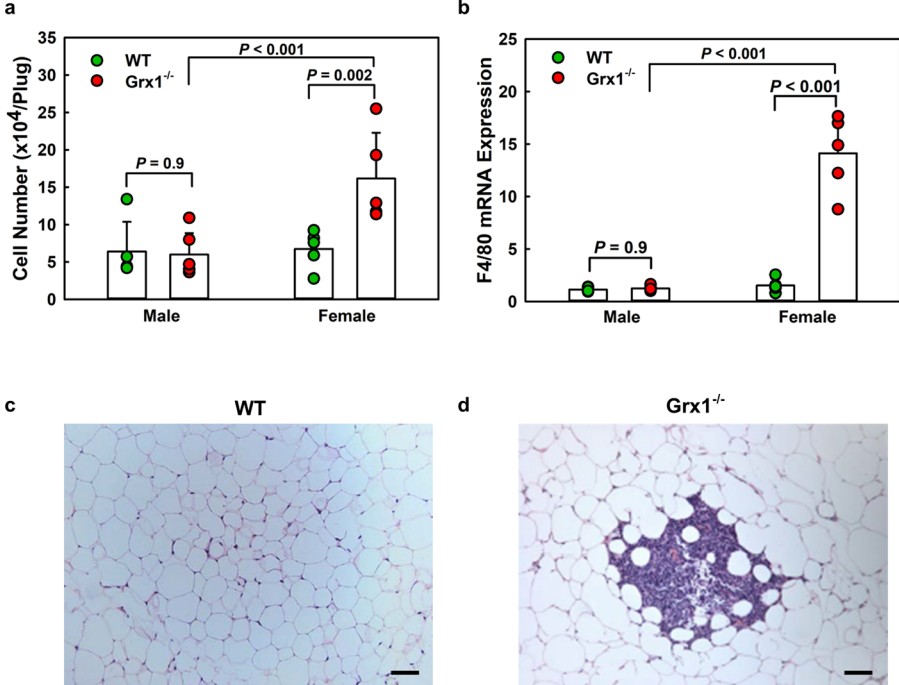

**Fig. 2 Assessment of monocyte chemotactic activity in vivo and macrophage recruitment into gonadal adipose tissues from aged male and female mice. a** Recruitment monocyte-derived macrophages into implanted Matrigel plugs loaded with. CP-1. **b** F4/80 mRNA expression in gonadal adipose tissues. **c, d** Representative images of gonadal adipose tissue section stained with H&E. Scale bar = 50 μm All data are expressed as mean ± S.D., $n$ = 5–6 mice per group for **a** and **b**. One-way ANOVA followed by Fisher's Least Significance Difference test was used to compare the mean values between experimental groups. *$P$ < 0.05; **$P$ < 0.01; ***$P$ < 0.001. Source data are provided as a Source Data file.

whether increased monocyte priming stimulated the formation of atherosclerotic lesions in aged female Grx1[-/-] mice. While none of the aged male Grx1[-/-] mice showed any significant lesion formation (Fig. 3a, c), all aged female Grx1[-/-] mice showed lipid accumulation in their aortas, evidence of early atherosclerotic lesions (Fig. 3b, c). Surprisingly, these lesions developed predominantly in the abdominal aorta rather than the aortic arch (Fig. 3d), the typical site in mouse models of atherosclerosis[22,23]. The onset of atherosclerosis in these mice was confirmed in the aortic root (Fig. 4a, b), where lesion development coincided, as expected, with the increased accumulation of CD68-positive monocyte-derived macrophages (Fig. 4c, d). Aortic root sections from these aged female Grx1[-/-] mice showed high levels of protein S-glutathionylation compared to comparable sections obtained from female WT mice (Fig. 4e, f). Protein S-glutathionylation was localized primarily to lesion areas and was particularly pronounced in CD68[+] macrophage-rich regions. Male mice showed no lesions in the aortic root (Fig. 4b).

**Grx1 deficiency in aged female mice promotes monocyte reprogramming by derepressing the expression of ROS and RNS generating enzymes and disrupting thiol redox homeostasis.** To determine whether Grx1 deficiency promotes the reprogramming of macrophages and whether this reprogramming is sexually dimorphic, we conducted gene-profiling experiments in peritoneal macrophages from all four groups of mice using a targeted RT-qPCR approach with custom-designed 384 TaqMan® Gene Expression Array Cards probing for 330 genes. These included 20 markers of proinflammatory activation and 25 markers associated with inflammation resolving macrophage phenotypes, as well as major transporters, transcription factors, antioxidants and ROS generators, signaling molecules and genes involved in key metabolic pathways (Fig. 5a and

Supplemental Table 2). Principal component analysis revealed that the gene signature of male Grx1[-/-] macrophages closely resembled that of male WT macrophages whereas female WT and Grx1[-/-] macrophages showed their own unique gene signature (Fig. 5b). In aged male mice, Grx1 deficiency resulted in the upregulation 12 out 330 genes ($P$ < 0.05), while only 7 genes were downregulated ($P$ < 0.05) (Fig. 5c). In contrast, in aged female mice, Grx1 deficiency upregulated the expression of 16 of the 330 genes whereas 15 were downregulated. Importantly, out of the 46 genes that showed significantly altered expression levels in Grx1[-/-] macrophages only 4 were common to both males and female mice (Fig. 5c). These findings confirm that Grx1 deficiency promotes the reprogramming of macrophages, and that this reprogramming is highly sexually dimorphic. This finding is in good agreement with the strong sexual dimorphic response we had observed on protein S-glutathionylation in macrophages isolated from male and female high-calorie diet-fed (HCD), atherosclerosis-prone LDLR[-/-] mice[6]. Macrophages from female LDLR[-/-] mice have 62% more S-glutathionylated proteins than macrophages isolated from male age-matched dyslipidemic LDLR[-/-] mice, even though male mice show both higher plasma cholesterol and higher triglyceride levels than females in response to the HCD[6]. Monocytes from female mice, therefore, appear to be more sensitive to thiol oxidative stress and protein S-glutathionylation in response to rising plasma lipid levels, which could explain the higher level of monocyte priming in aged female Grx1[-/-] mice compared to male mice (Fig. 2a). However, the mechanism underlying this sexual dimorphic (thiol) oxidative stress response to nutrient stress was not known.

The majority of gene expression changes in both male and female Grx1[-/-] macrophages were modest (<2-fold), with the notable exception of genes encoding ROS and RNS generating enzymes (Fig. 5a). We observed a sexually dimorphic

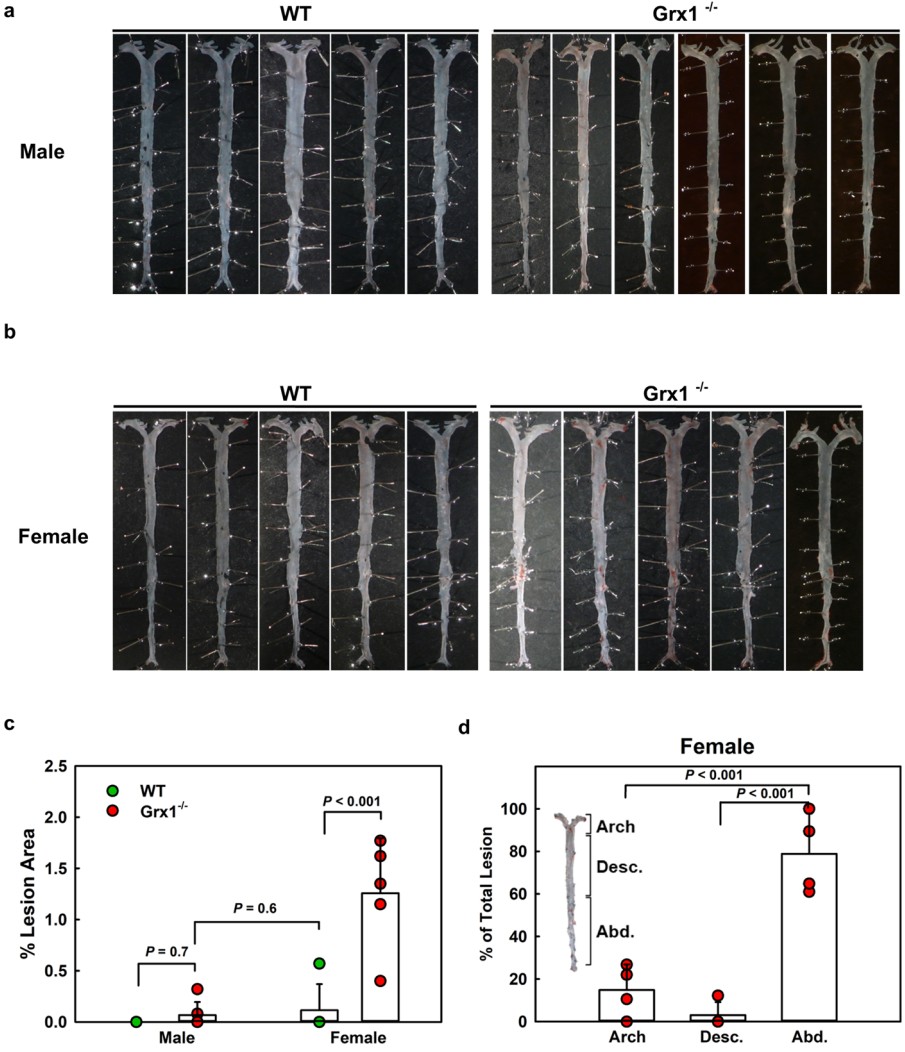

**Fig. 3 Grx1 deficiency in chow-fed aged female Grx1⁻/⁻ mice promotes atherosclerotic lesion formation. a, b** Images of Oil Red O-stained aorta from male and female 18-month-old Grx1⁻/⁻ Mice and age-matched male and female C57BL/6 J mice fed a normal chow. **c** Quantitation of aortic lesion area. $n = 5$–6 mice per group. **d** Distribution of lesions between aortic arch, descending aorta and abdominal aorta in female Grx1⁻/⁻ mice, $n = 5$ mice per group. All data are expressed as mean ± S.D. One-way ANOVA followed by Fisher's Least Significance Difference test was used to compare the mean values between experimental groups. Source data are provided as a Source Data file.

*de*repressing effect of Grx1 deficiency on ROS and RNS generators in both male and female macrophages. In male mice, Grx1 deficiency upregulated NADPH oxidase 1 (NOX1), neuronal (NOS1) and inducible NO synthase (NOS2) (Supplemental Fig. 4a–c), whereas Grx1 deficiency in macrophages from female mice upregulated primarily NADPH oxidase 4 (NOX4) and endothelial NO synthase (NOS3) and to a lesser extent neuronal NOS synthase (NOS1, Fig. 5d, e, Supplemental Fig. 4b). Most notably, compared to 18-month-old male WT mice, macrophages from 18-month-old Grx1⁻/⁻ male mice showed only a 42-fold induction of NOX4 whereas female Grx1⁻/⁻ macrophages showed a 352-fold induction (Fig. 5d). This inducible source of $H_2O_2$ was only recently identified in human monocyte-derived macrophages and was shown to mediate oxidized LDL-induced macrophage death[24]. We went on to show that nutrient stress induces Nox4 in human THP-1 monocytes, that over-expression on NOX4 is sufficient to promote monocyte priming and that monocyte priming by nutrient stress is prevented by siRNA-mediated Nox4 knockdown[5]. Therefore, the derepression and dramatic overexpression of NOX4 in female Grx1⁻/⁻

macrophages appears to be sufficient to prime monocytes and macrophages in female Grx1⁻/⁻ mice (Fig. 2a) and accelerate the recruitment of monocyte-derived macrophages into aortic lesions (Fig. 4c, d) and adipose tissues (Fig. 2b).

Interestingly, in addition to NOX4, female Grx1⁻/⁻ macrophages also showed a dramatic 601-fold induction of endothelial nitric oxide synthase (NOS3,) and 2.8-fold induction of ARG2 (Supplemental Fig. 4d) poising female but not male Grx1⁻/⁻, Fig. 5e) macrophages for increased peroxynitrate production. ARG2 competes with NOS for L-arginine and, under conditions of L-arginine restriction, NOS become uncoupled and generate superoxide at the expense of NO[25,26]. Indeed, we found strong nitrotyrosine staining in the aortic roots of aged female Grx1⁻/⁻ mice, a marker of peroxynitrate-mediated protein damage, whereas Grx1 deficiency had no effect on nitrotyrosine formation in aged male mice (Fig. 6). We also found no nitrotyrosine staining in the aortic roots of WT male or female mice. In macrophages, NOS3 appears to regulate cellular activation and proinflammatory responses[27]. How increased expression of NOS3 contributes to monocyte priming and macrophage dysfunction and the dramatic

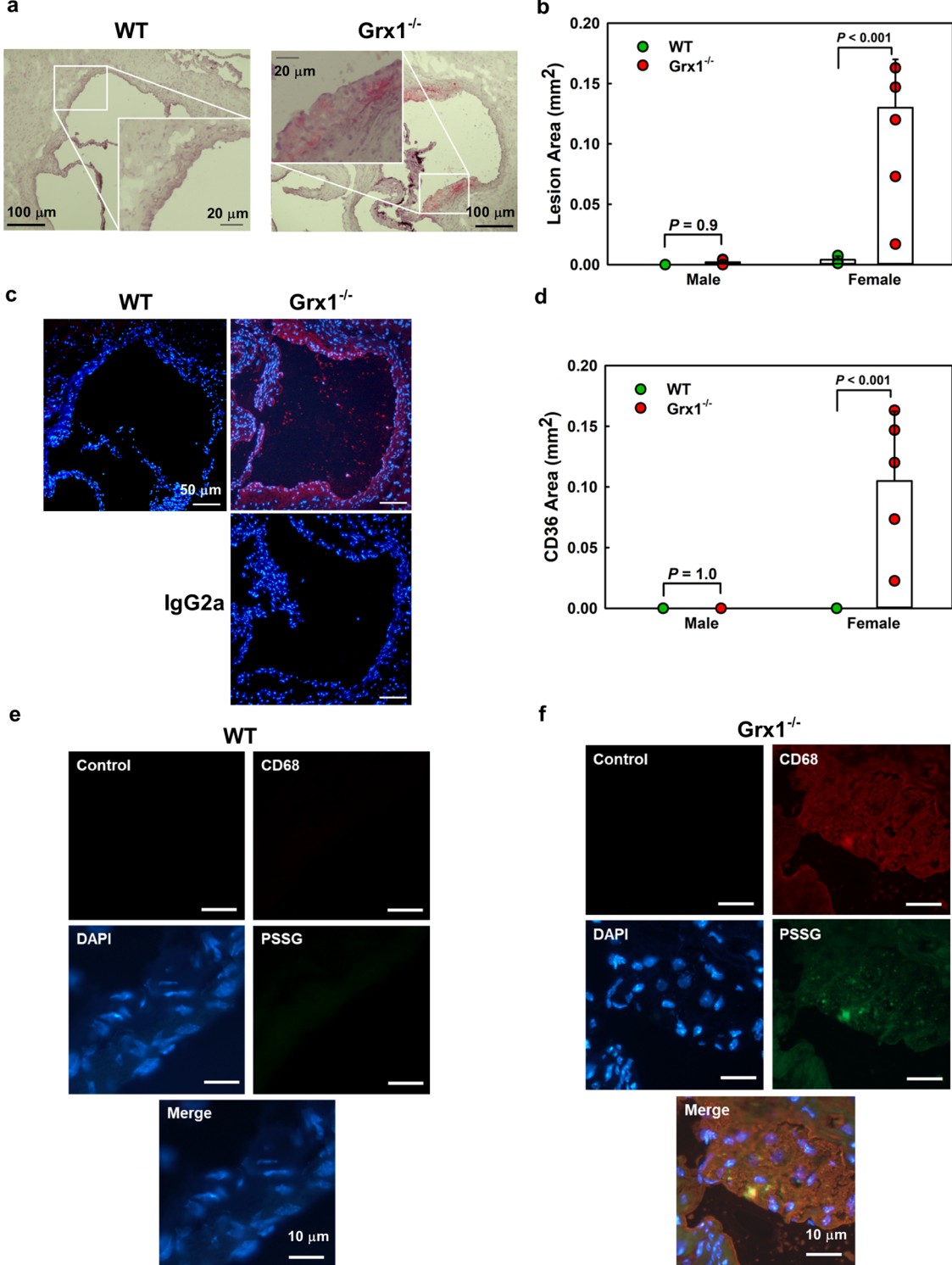

effects of Grx1 deficiency in aged female mice is not clear, but reprogramming of the proteome in response to peroxynitrite-mediated protein *S*-glutathionylation is a likely mechanism[28,29].

Surprisingly, we found no evidence for enhanced peroxinitrite formation in aged male Grx1[-/-] mice (Fig. 6) despite the induction of NOX1, a generator of superoxide required for peroxinitrate formation, and NOS1 and NOS2, which could deliver the necessary nitric oxide. However, with the exception of NOX4, ROS formation by NADPH oxidases is not regulated at the transcriptional level[30]. Thus, elevated mRNA levels do

not necessarily lead to increased NOX activity and ROS formation. On the other hand, NOX4 expression was increased over 350-fold in aged female Grx1[-/-] mice (Fig. 5d), but is unlikely to account for the peroxinitrate formation we detected as NOX4 predominately generates $H_2O_2$, not superoxide[31]. While we clearly showed that peroxinitrate is formed in lesions of aged female Grx1[-/-] mice, we can only speculate on the underlying mechanism and the source of superoxide required. It appears that the major difference between aged male and female Grx1 KO mice is the massive induction of NOS3 in aged

**Fig. 4 Grx1 deficiency in chow-fed female Grx1$^{-/-}$ mice promotes atherogenesis and drives macrophage recruitment and protein S-glutathionylation in atherosclerotic lesions. a** Representative images of Oil Red O-stained sections from the aorta root of an 18-month-old female Grx1$^{-/-}$ mice and an age-matched male and female C57BL/6 J mouse (WT) maintained on a normal chow diet. Scale bar = 100 μm. Insert: Scale bar = 20 μm. **b** Quantitation of lesion area in the aortic root of male and female Grx1$^{-/-}$ mice and age-matched male and female C57BL/6 J mice (WT) maintained on a normal chow diet for 18 months. n = 5–6 mice per group. **c** Representative images of aorta root sections stained with either rat IgG2a (Control) or a macrophage-specific antiserum against CD68 (CD68) from an 18-month-old female Grx1$^{-/-}$ mice and an age-matched female C57BL/6 J mouse (WT). Scale bar = 50 μm. **d** Macrophage content of atherosclerotic lesions in the aorta root of 18-month-old female Grx1$^{-/-}$ mice and age-matched male and female C57BL/6 J mice (WT) maintained on a normal chow diet for 18 months. n = 5–6 mice per group with male and female Grx1$^{-/-}$ mice and age-matched male and female C57BL/6 J mice (WT). **e,f** Representative images of aorta root sections labeled in situ for S-glutathionylated proteins (green), stained with either rat IgG2a (Control, red) or a macrophage-specific antiserum against CD68 (red) and DAPI (blue) to identify nuclei. Colocalization of protein S-glutathionylation and CD68-positive areas are shown in yellow. Magnification: 60x; Scale bar = 10 μm. All data are expressed as mean ± S.D. One-way ANOVA followed by Fisher's Least Significance Difference test was used to compare the mean values between experimental groups. Source data are provided as a Source Data file.

---

female Grx1$^{-/-}$ mice, which greatly exceeded the induction of NOS enzymes in aged male Grx1$^{-/-}$ mice (Fig. 5e versus Supplemental Fig. 4b, c), and the induction of ARG2 (Supplemental Fig. 4d). The combined induction of NOS3 and ARG2 in itself could account for the formation of peroxynitrite in aged female Grx1$^{-/-}$ mice[25,26]. Alternatively, although CYBB, the gene for NOX2, was not induced in either aged male or female Grx1$^{-/-}$ mice, this NADPH oxidase is the major source of superoxide in macrophages[30] and may also provide the superoxide needed for peroxinitrite formation in aged female Grx1$^{-/-}$ mice.

Taken together, our data reveal a previously unknown, sexually dimorphic negative feedback mechanism in macrophages between the antioxidative activity of Grx1 and the expression of ROS and RNS generators. The combined derepression and upregulation of NOX4, NOS3, and ARG2 in macrophages from aged female Grx1$^{-/-}$ mice appear to convert these primed and reprogrammed monocyte-derived macrophages into proinflammatory, proatherogenic and proobesogenic phenotypes. This phenotypic change alone may account for both the onset of atherogenesis and accelerated weight gain we observed in aged female Grx1$^{-/-}$ mice despite the absence of any significant diet-induced nutrient stress (Fig. 1d, f, Supplemental Fig. 10). This mechanistic link we have identified between Grx1 and atherosclerosis is supported by gene-disease association data from the Comparative Toxicogenomics Database (CDT) in the Harmonizome database, demonstrating a strong association of Glrx1 with both atherosclerosis and heart disease[13]. However, in mice, Grx1 deficiency unmasked obesity and cardiovascular risk potential only in aging females, but not in male mice. Grx1 appears to play a unique anti-obesogenic and atheroprotective role in reproductively senescent female mice (Supplemental Fig. 10).

The onset of rapid weight gain at 7–8 months in chow-fed aging female Grx1$^{-/-}$ mice coincides with the onset of irregular cycling and reproductive senescence—aging female mice show declining estrogen levels but do not undergo true menopause[32–34]. Data from our redox proteomics studies suggest that monocytes and macrophages from female mice are fundamentally more sensitive to nutrient stress-induced protein S-glutathionylation, priming and dysfunction[6]. Macrophage dysfunction is a rate-limiting process in the development of obesity and cardiovascular diseases, and female Grx1$^{-/-}$ mice do not exhibit any accelerated weight gain or atherosclerotic lesion formation until after reaching the age of irregular cycling and declining estrogen levels. Protection conveyed by estrogen may have prevented monocyte priming and dysfunction in young female Grx1$^{-/-}$ mice. Our findings suggest that in female macrophages the disruption of thiol redox homeostasis associated with reproductive senescence is a mechanism that may contribute

to the higher rates of obesity and CVD observed in postmenopausal women[35].

**Hematopoietic Grx1 deficiency accelerates atherosclerosis and weight gain by sensitizing blood monocytes to HCD-induced priming and dysfunction.** Recruitment of monocyte-derived macrophages is a common step required for AT inflammation and obesity as well as for atherogenesis[10,11,36–38]. We therefore examined whether Grx1 deficiency in myeloid cells is sufficient to drive weight gain and atherosclerosis in reproductively healthy female mice. To this end, we conducted bone marrow (BM) transplantation (BMT) studies in male and female atherosclerosis-prone LDLR$^{-/-}$ mice using 10-week-old WT or Grx1$^{-/-}$ male and female mice, respectively, as BM donors. Recipient mice were fed an HCD to induce atherogenesis and weight gain. Only 4 weeks after initiating HCD-feeding, female Grx1$^{-/-}$ BM recipients (Grx1$_{Leuko}^{-/-}$) showed a significant acceleration in body weight gain, and after 20 weeks on the HCD, Grx1$_{Leuko}^{-/-}$ were 14.6% heavier than control mice (Fig. 7a). After 6 weeks on HCD, female Grx1$_{Leuko}^{-/-}$ mice also showed significantly elevated fasting blood glucose levels compared to control mice, indicating these mice were becoming insulin resistant (Fig. 7b). The weight gain appears to occur in the adipose tissue and the liver, whereas the kidneys were unaffected (Supplemental Fig. 5a–c). Yet, neither total plasma cholesterol and triglyceride levels nor the lipoprotein profiles were significantly different between the two groups (Fig. 7c–e), suggesting that other, possible non-metabolic factors were driving the accelerated weight gain. Indeed, the earliest changes we observed were the dramatic increase in macrophage content and appearance of crown-like structures in the AT of Grx1$_{Leuko}^{-/-}$ mice, which occurred after only 6 weeks on HCD (Supplemental Fig. 6a, b). Increased macrophage infiltration (Supplemental Fig. 6c), however, was not due to increased blood monocyte counts (Supplemental Table 3).

As predicted, we also observed accelerated atherogenesis in female Grx1$_{Leuko}^{-/-}$ mice, both in the aorta (Fig. 8a, b) as well as the aortic root (Fig. 8c, d), where significant differences in lesion size were detectable as early as 6 weeks after initiating the HCD. After 20 weeks, atherosclerotic plaques in the aorta of Grx1$_{Leuko}^{-/-}$ mice were 29% larger than those found in recipients of WT BM and 17% larger in the aortic root. Lesions size in the aorta was increased similarly across the entire aortic root (Fig. 8e), suggesting that changes in humoral factors rather than an alteration in the vessel are likely to be responsible for the accelerated atherogenesis.

In contrast, transplanting BM from 10-week-old male Grx1$^{-/-}$ mice into male LDLR$^{-/-}$ mice (Supplemental Fig. 7a) had no effect on HCD-induced atherogenesis (Supplemental Fig. 7b) and did not accelerate weight gain (Supplemental Fig. 8). These findings suggest that the susceptibility of aged female Grx1$^{-/-}$ mice to

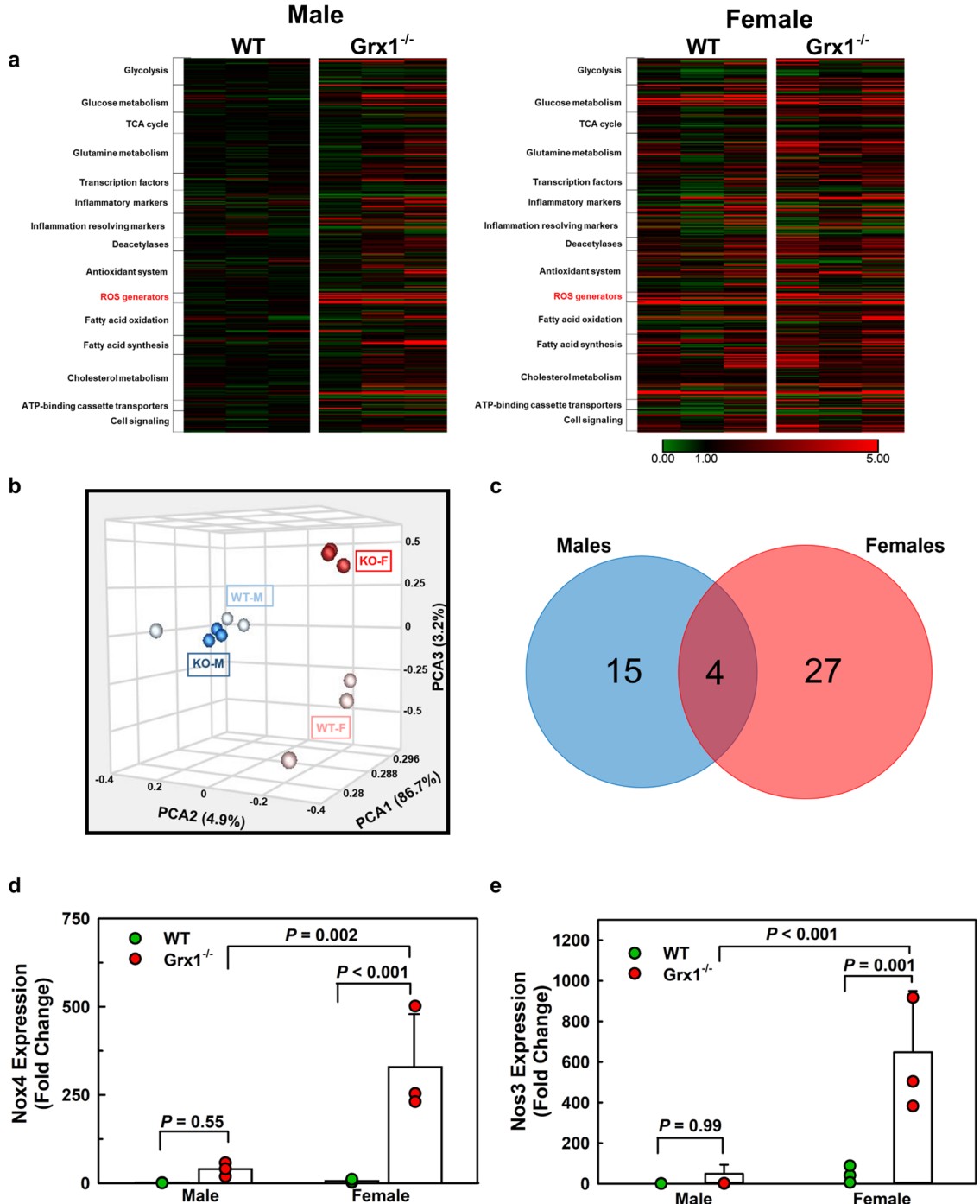

**Fig. 5 Macrophage Reprogramming in Aged Grx1-Deficient Mice. a** Heatmap representing normalized gene expression in peritoneal macrophages isolated from male and female Grx1$^{-/-}$ mice and age-matched male and female C57BL/6 J mice (WT) maintained on a normal chow diet for 18 months. **b** Principal component analysis (PCA) of variances in the mRNA expression in peritoneal macrophages of male (**blue**) and female (**red**) WT and Grx1$^{-/-}$ mice (KO). **c** Venn diagram displaying differentially expressed genes between peritoneal macrophages from male (**blue**) and female mice (**red**). **d** Derepession of NOX4 expression by Grx1 deficiency in peritoneal macrophages from 18-month-old female but not male mice. **e** Derepession of NOS3 expression by Grx1 deficiency in peritoneal macrophages from 18-month-old female but not male mice. All data are expressed as mean ± S.D, $n = 3$. One-way ANOVA followed by Fisher's Least Significance Difference test was used to compare the mean values between experimental groups. Source data are provided as a Source Data file.

develop atherosclerotic plaques is conferred by hematopoietic cells rather than loss of estrogen or other hormonal factors as young female mice with normal estrogen levels and hematopoietic Grx1 deficiency also show accelerated atherogenesis and increased weight gain, yet HCD-fed male LDLR$^{-/-}$ mice were unaffected by hematopoietic Grx1 deficiency.

Furthermore, in contrast to female BM recipients (Fig. 9a), HCD-induced loss of MKP-1 activity in blood monocytes of male LDLR$^{-/-}$ BM recipients was not further enhanced by hematopoietic Grx1 deficiency (Supplemental Fig. 7c). This suggests that the rate-limiting event in HCD-induced inactivation of MKP-1 in monocytes and macrophages is the oxidation

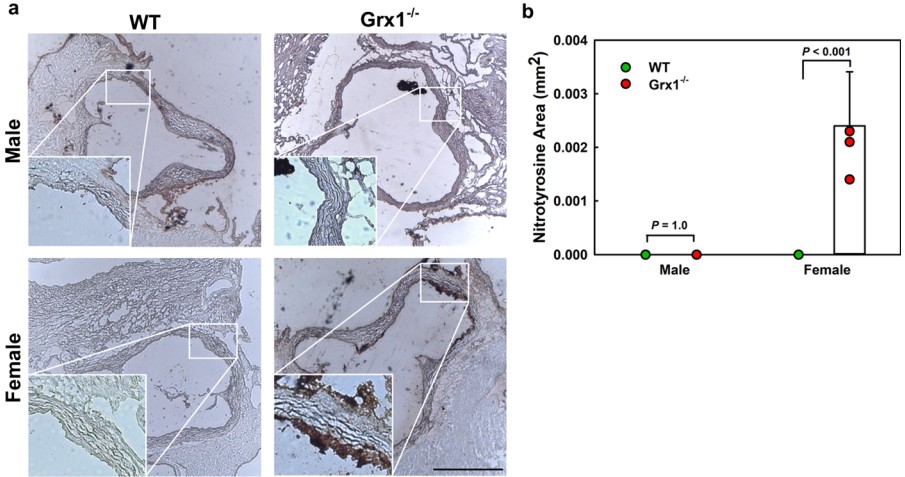

**Fig. 6 Hematopoietic Grx1 deficiency promotes formation of nitrotyrosine in the aortic root of aged female but not male mice. a** Representative images of aorta root sections stained with an antiserum against nitrotyrosine from 18-month-old male and female WT and Grx1[-/-] mice maintained on a normal chow diet for 18 months. Scale bar: 500 μm. Inserts: 100x magnification. **b** Nitrotyrosine levels measured in the aortic root of male and female WT and Grx1[-/-] mice maintained on a normal chow diet for 18 months. All data are expressed as mean ± S.D, n = 5–6 per group, 8 aortic root sections per mouse. Source data are provided as a Source Data file.

and *S*-glutathionylation of its catalytic cysteine rather than the (impaired) reduction, i.e. deglutathionylation, and reactivation of MKP-1[3]. MKP-1 is a master regulator of macrophage function and controls numerous signaling pathways, cellular functions as well macrophage activation states[39]. The differences in MKP-1 activity in response to HCD-feeding may at least in part explain the differences in macrophage "phenotype" between male and female Grx1[-/-] mice and thus their propensity to drive atherogenesis.

Our findings also suggest that the sex difference observed in aged Grx1[-/-] mice is of hematopoietic origin and imply that differences in epigenetic programming of macrophages between male and female mice rather than hormonal control underlie the observed sexual dimorphism. This conclusion is supported by the fact that the purified peritoneal macrophages we used for the gene profiling data, were cultured for 2 days in the absence of sex hormones prior to their analysis, ruling out any direct effects of sex hormones. This, of course, does not rule out that sex hormones play an important role in determining the epigenetic landscapes of male and female macrophages and their precursors. In either case, our data suggest that in female macrophages the disruption of thiol redox homeostasis combined with reproductive senescence is a mechanism that may contribute to the higher rates of obesity and CVD observed in postmenopausal women[35].

To address the mechanism underlying accelerated atherogenesis in female Grx1[Leuko][-/-] mice, we examine whether their blood monocytes were primed by Grx1 deficiency beyond the priming and reprogramming induced by the HCD alone. To this end, we again used our Matrigel plug assay[8] to determine the chemotactic activity of blood monocytes in these mice. Indeed, monocytes in Grx1[Leuko][-/-] mice were highly primed for accelerated chemotaxis as we detected 2.2-fold more macrophages in MCP-1-loaded Matrigel plugs of Grx1[Leuko][-/-] mice than in plugs removed from recipients of WT BM (Fig. 8f); this, despite the fact that monocytes in HCD-fed LDLR[-/-] mice are already primed and show a 3-fold increase in chemotactic activity compared to monocytes from LDLR[-/-] mice fed a low-calorie maintenance diet (MD)[7]. This increase in monocyte chemotactic activity in female Grx1[Leuko][-/-] mice may explain the accelerated accumulation of macrophages we observed in both the adipose tissues (Supplemental Fig. 6) as well as in the

atherosclerotic plaques of these mice (Fig. 8g, h), as neither total plasma lipid levels nor lipoprotein distributions differed between the two groups of mice (Fig. 7c–e).

Our data from the aged Grx1[-/-] mice suggest that in the absence of HCD-induced nutrient stress, Grx1 deficiency alone should be sufficient to promote the conversion of female monocytes and macrophages into the same primed, hyper-chemotactic proatherogenic phenotypes we observe in HCD-fed mice. To test this hypothesis, we used a modified Boyden chamber to measure MCP-1-induced chemotaxis in purified peritoneal macrophages isolated from female WT and Grx1[-/-] mice. Indeed, Grx1[-/-] macrophages showed a 2-fold higher chemotactic activity than WT macrophages (Fig. 9b, Untreated). In agreement with our in vivo chemotaxis data (Fig. 8f), Grx1-deficiency also further enhanced nutrient stress-induced priming and increased macrophages chemotaxis by 28% (Fig. 9b, HG + LDL), supporting a protective role for Grx1 in monocytes and macrophages against nutrient stress-induced dysfunction (Supplemental Fig. 10).

**Macrophage-restricted overexpression of Grx1 protects blood monocytes against HCD-induced priming, prevents HCD-induced expression of ROS and RNS generating enzymes in macrophages, and reduces the severity of atherosclerosis**. Our data suggest that Grx1 plays an essential role in protecting monocytes and macrophages from reprogramming and dysfunction induced by aging or HCD-triggered nutrient stress. To determine if macrophage Grx1 activity is not only necessary but also sufficient to prevent monocytes and macrophages reprogramming and thus to protect against atherogenesis, we conducted gene transfer experiments in LDLR[-/-] mice to overexpress Grx1 in a macrophage-restricted manner. We generated bicistronic lentiviral vectors that express either EGFP or Grx1 and EGFP under the control of the CD68 promoter[40,41].

BM was harvested from 10-week-old male and female C57/BL/6 J mice, hematopoietic progenitor cells were isolated and transduced with the lentiviral vectors and transplanted into 10-week-old male and female LDLR[-/-] mice, respectively. The presence of LDL receptors on bone marrow cells does not affect atherosclerotic lesion size when transplanted into LDLR[-/-] mice[42]. Recipient mice were fed a HCD for 20 weeks to induce atherogenesis. In addition, a group of control LDLR[-/-] mice were fed a low-calorie maintenance diet (MD) for 20 weeks. BMDM

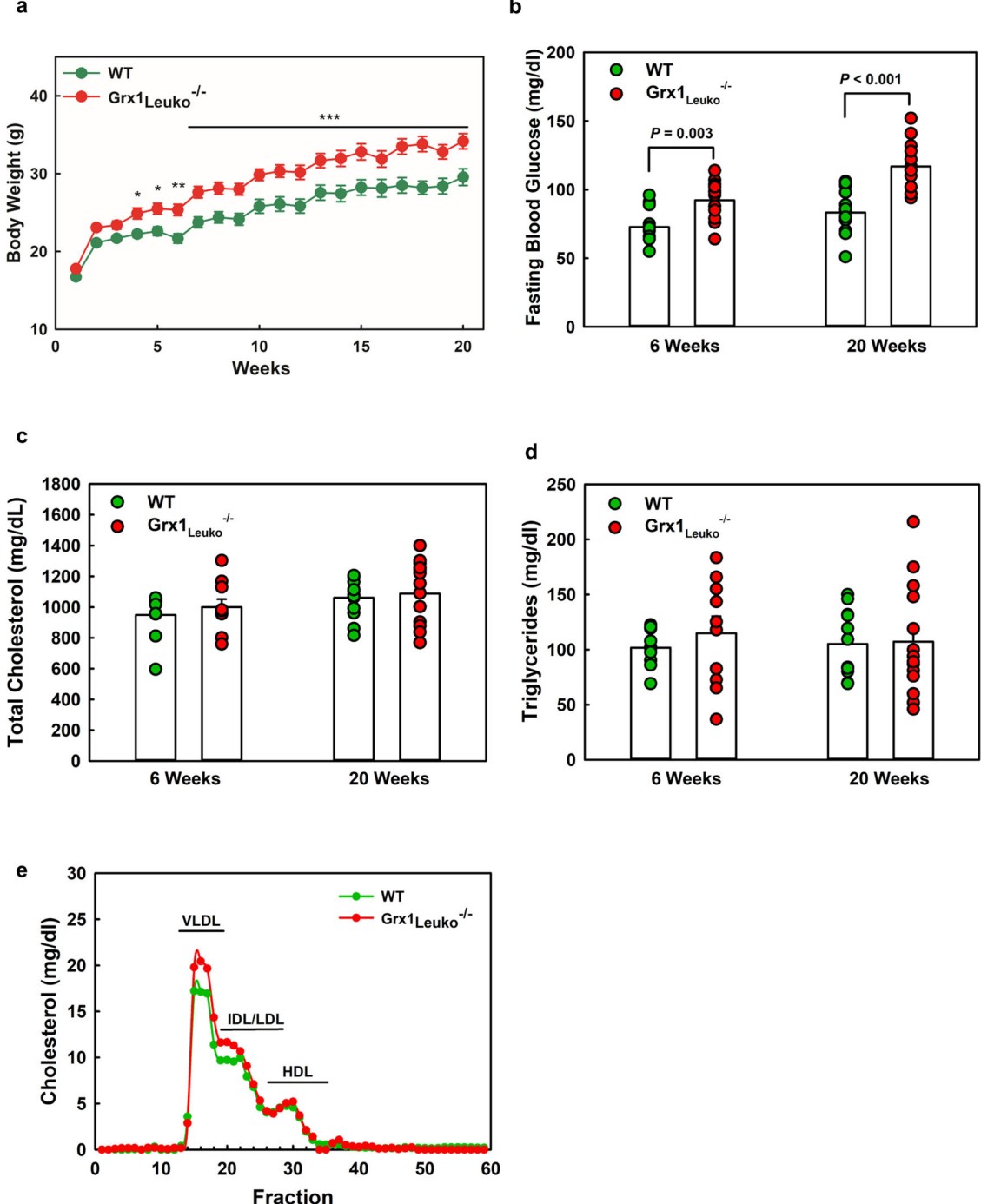

**Fig. 7 Hematopoietic Grx1 deficiency accelerates body weight gain in reproductively healthy female LDLR$^{-/-}$ mice fed a high-calorie diet. a** Body weights. $n = 14$ per group. **b** Fasting blood glucose levels. $n = 14$ per group. **c** Total cholesterol levels. $n = 14$ per group. **d** Total plasma triglyceride levels. $n = 14$ per group. **e** Lipoprotein profiles of LDLR$^{-/-}$ (WT) and LDLR$^{-/-}$Grx1$_{Leuko}$$^{-/-}$ mice (Grx1$_{Leuko}$$^{-/-}$) from pooled plasma samples; $n = 14$ per group. All data are expressed as mean ± S.E. One-way ANOVA followed by Fisher's Least Significance Difference test was used to compare the mean values between experimental groups. Source data are provided as a Source Data file.

were isolated from all mice and subjected to gene profiling by qPCR. Male BM recipients transduced with Grx1-carrying lentiviruses showed a 6.4-fold increase in Grx1 expression whereas females showed only a 3.8-fold increase in Grx1 expression compared to BMDM from the respective MD-fed controls (Fig. 10a). Overexpression of EGFP in macrophages from HCD-fed LDLR$^{-/-}$ mice had no effect on Grx1 mRNA levels (Fig. 10a). HCD-feeding inhibited MKP-1 activity by 51% in male and 66% in female BMDM, indicating increased protein

S-glutathionylation in these cells. We showed previously that one of the targets of HFD-induced protein S-glutathionylation is Grx1[28]. S-glutathionylated Grx1 is enzymatically inactive[43], suggesting that HCD-induced nutrient stress promotes a state of Grx1 deficiency in macrophages, which is sufficient to induce – or derepress — the expression of ROS and RNS generating enzymes in a sexually dimorphic manner (Fig. 10c–h). In male macrophages, HFD-induced nutrient stress-induced primarily CYBB and NOX4 as well as NOS1, 2 and 3, whereas female

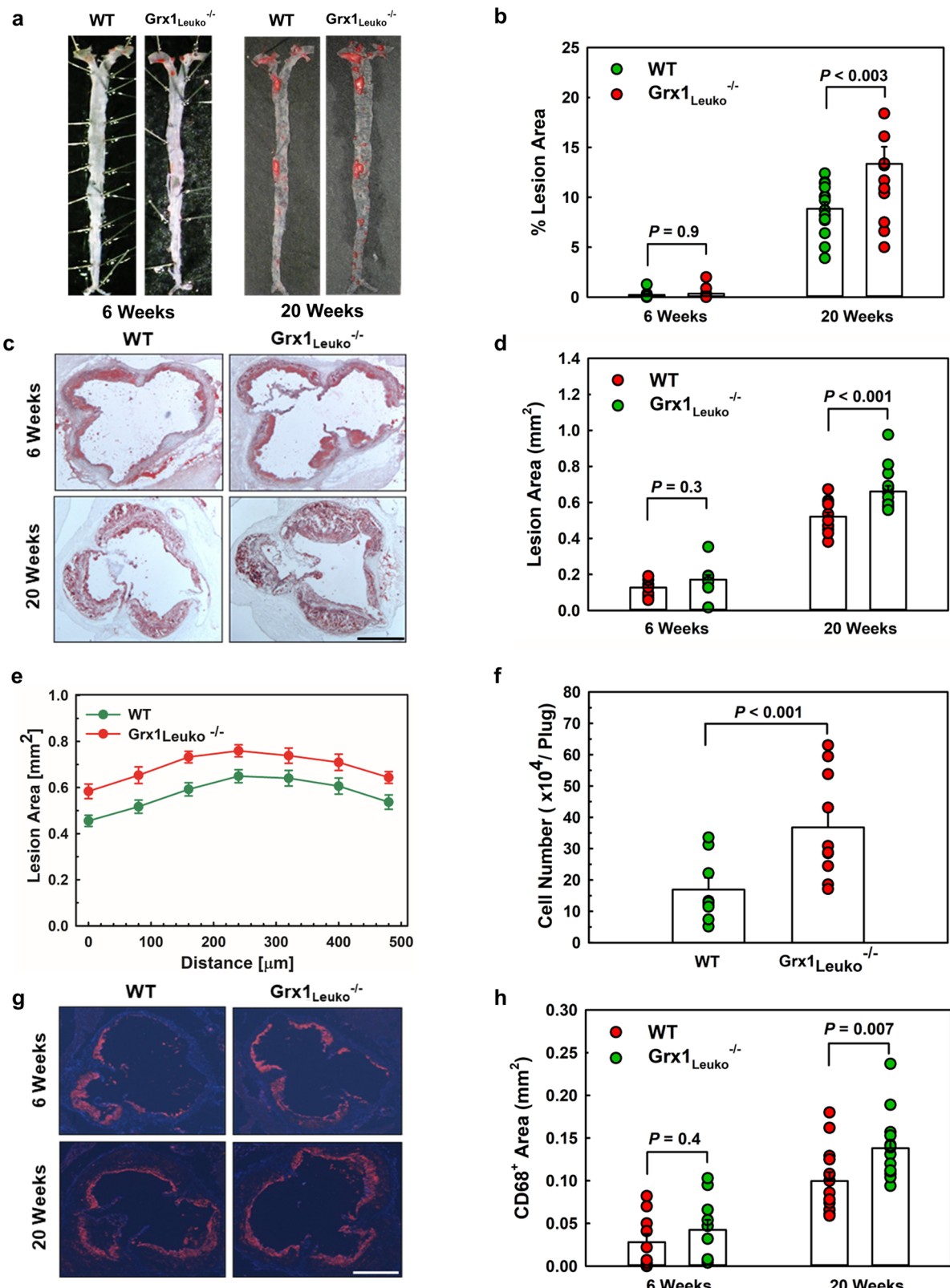

macrophages showed no increase in NOX but a robust expression of NOS1, 2 and 3. These NOX and NOS profiles differ from those seen in macrophages isolated from aged Grx1$^{-/-}$ mice, indicating that the epigenetic landscapes of nutrient-stressed macrophages from young, LDLR$^{-/-}$ mice are different.

HCD-feeding induced atherogenesis in both males and females, with a 62% greater plaque burden observed in female

mice (Supplemental Fig. 10). Overexpression of Grx1 in macrophages not only reversed the induction of NOX, NOS, and ARG2, eliminating the sex differences in ROS and RNS generators, but also reduced atherosclerosis by 39% in male and by 35% in female mice, again, eliminating the sex difference we observed in HCD-fed mice overexpressing only EGFP (Supplemental Fig. 10). These findings suggest that increasing Grx1

**Fig. 8 Hematopoietic Grx1 deficiency promotes atherosclerosis in reproductively healthy female LDLR$^{-/-}$ mice by enhancing recruitment of monocyte-derived macrophages into atherosclerotic lesions. a** Representative images of Oil Red O-stained aortas from female LDLR$^{-/-}$Grx1$_{Leuko}$$^{-/-}$ mice (Grx1$_{Leuko}$$^{-/-}$) and age-matched female LDLR$^{-/-}$ mice (WT) fed a HCD. **b** Quantification of aortic lesion area. $n = 14$ per group. Week 6: $n = 10$; Week 20: $n = 14$. **c** Representative images of Oil Red O-stained sections from the aorta roots of HCD-fed female LDLR$^{-/-}$Grx1$_{Leuko}$$^{-/-}$ (Grx1$_{Leuko}$$^{-/-}$) and age-matched female LDLR$^{-/-}$ mice (WT). Scale bar = 500 μm. **d** Quantitation of lesion area in the aortic root. Week 6: $n = 10$; Week 20: $n = 14$ per group. **e** Distribution of lesion size across a 500 μm segment of the aortic root beginning at the aortic valve. $n = 14$ mice per group. **f** Recruitment monocyte-derived macrophages into implanted Matrigel plugs loaded with. MCP-1. $n = 14$ mice per group. **g** Representative images of aortic root sections stained with a macrophage-specific antiserum against CD68. Scale bar = 500 μm. **h** Quantitation of macrophage content in atherosclerotic lesions in the aortic root. Week 6: $n = 10$; Week 20: $n = 14$ mice per group. All data are expressed as mean ± S.E. One-way ANOVA followed by Fisher's Least Significance Difference test was used to compare the mean values between experimental groups. Source data are provided as a Source Data file.

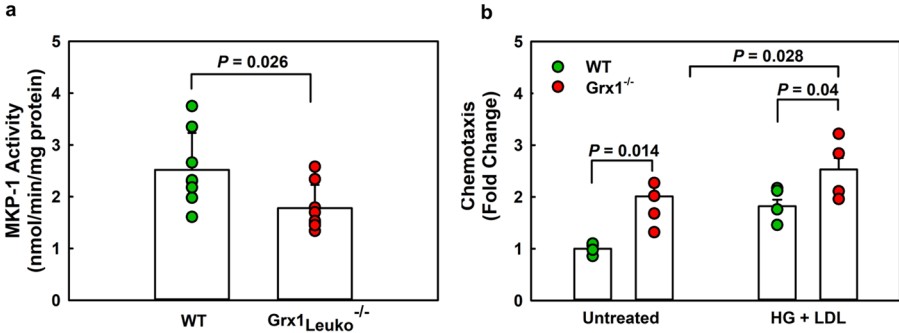

**Fig. 9 Hematopoietic Grx1 deficiency exacerbates nutrient stress-induced monocyte dysfunction. a** MKP-1 activity in purified peritoneal macrophages isolated from HCD-fed female LDLR$^{-/-}$Grx1$_{Leuko}$$^{-/-}$ mice (Grx1$_{Leuko}$$^{-/-}$) and age-matched female LDLR$^{-/-}$ mice (WT). $n = 15$ mice per group. **b** Chemotaxis of purified peritoneal macrophages isolated from female Grx1$_{Leuko}$$^{-/-}$ and age-matched C57BL/6 mice (WT). $n = 4$ mice per group. All data are expressed as mean ± S.D. One-way ANOVA followed by Fisher's Least Significance Difference test were used to compare the mean values between experimental groups. Source data are provided as a Source Data file.

activity may represent a therapeutic strategy for the prevention and treatment of atherosclerosis. We recently reported that inhibition of myeloid HDAC2 upregulates Grx1 expression, improves protein thiol redox state, protects against HCD-induced monocyte dysfunction and reduces atherosclerosis by 31%[44]. Unfortunately, benefits of HDAC2 inhibition were only observed in male mice and deletion of HDAC2 had no effect in females.

A common feature of macrophages from aged Grx1$^{-/-}$ mice and HCD-fed atherosclerosis prone LDLR$^{-/-}$ mice is the induction of ARG2 in female but not in male macrophages. In conjunction with the robust induction of NOS1, 2, and 3 in female macrophages, we predicted that these cells would show increased peroxynitrute formation due to NOS uncoupling. Indeed, we observed 28% more nitrotyrosine staining in the lesions of female mice compared to their male littermates (Fig. 10j), which may account for their 62% higher plaque burden (Supplemental Fig. 10). Multiple groups have reported that female LDLR$^{-/-}$ mice fed a HCD developed significantly more atherosclerosis over time than age-matched male LDLR$^{-/-}$ mice, although this sex difference appears to depend on the age of the mice and the vascular bed analyzed[43,44]. The underlying mechanisms are not fully understood, but our data suggest that the sexual dimorphic reprogramming of macrophages, especially the differential expression of NOX, NOS and ARG2 triggered by HCD-induced loss of Grx1 activity may contribute to this important sex difference.

In summary, we identified the thiol transferase Grx1 as a critical component in the defense of blood monocyte and monocyte-derived macrophages against nutrient stress-induced dysfunction and reprogramming, providing a link between HCD-induced (thiol) oxidative stress and atherosclerosis and obesity. We also uncovered a sexually dimorphic mechanistic link between Grx1 activity, obesity and the atherogenesis that is mediated by the differential expression of ROS and RNS-

generating enzymes in male and female macrophages. This macrophage-dependent mechanism may contribute to the well-established differences in cardiovascular risk between men and women, and to the elevated obesity and cardiovascular risk among postmenopausal women.

## Methods

**Compliance statement.** All studies comply with the relevant ethical regulations and were performed in accordance with the guidelines and regulations of and with the approval of the University of Texas at San Antonio Health and Wake Forest School of Medicine Institutional Animal Care and Use Committees.

**Animals and diets.** LDLR$^{-/-}$ (B6.129S7-*Ldlr*$^{tm1Her}$/J, stock number 002207) and C57BL/6 J (stock number 000664) were obtained from Jackson Laboratory. Grx1$^{-/-}$ mice were kindly provided by Dr. Yvonne Janssen-Heininger, University of Vermont, with kind permission from Dr. Ye-Shih Ho, Wayne State University. The mice were backcrossed into the C57BL/6 J genetic background for more than 10 generations. All mice were maintained in colony cages on a 12-h light/12-h dark cycle and fed a normal mouse laboratory diet unless otherwise stated. For the aging study, C57BL/6 J (wild-type; 5 males and 5 females) and Grx1$^{-/-}$ mice (6 males and 5 females) were fed chow diet (5.5% fat wt/wt, Prolab® RMH 3000 5P00, LabDiet) for 18 months. Body weights and fasting glucose levels were measured every 4 weeks. For the bone marrow (BM) transplantation experiments, 4 weeks after the transplantation, male and female BM-recipient mice were switched to a high-calorie diet (HCD; 21% milk fat wt/wt and 0.2% cholesterol wt/wt. diet no. F55440, Bio-Serv) for 6 or 20 weeks. Body weights were measured weekly and fasting blood glucose every 3 weeks using a glucometer. All studies were performed in accordance with the guidelines and regulations of and with the approval of the Institutional Animal Care and Use Committees.

**Lentiviral vector construction and generation of transgenic mice with macrophage-restricted EGFP or Grx1 and EGFP overexpression.** The CD68 promoter construct, which contains the 83-bp first intron (IVS-1) of the human CD68 gene, was kindly provided by Dr. David Greaves, University of Oxford. CD68-IRES-EGFP in pcDNA3.1 vector was digested *ClaI*/*NotI* and inserted into the corresponding site in the pRRL.cPPT.PGK-GFP.WPRE.Sin-18 backbone (Addgene), replacing the hPGK promoter and EGFP region. Lentiviruses were produced by transfecting the vector into HEK293T cells (ATCC, CRL-3216) using a lentivirus packaging kit (Origene) and then concentrating the viruses with the Lenti Concentrator (Origene). Bone marrow (BM) cells were harvested from the

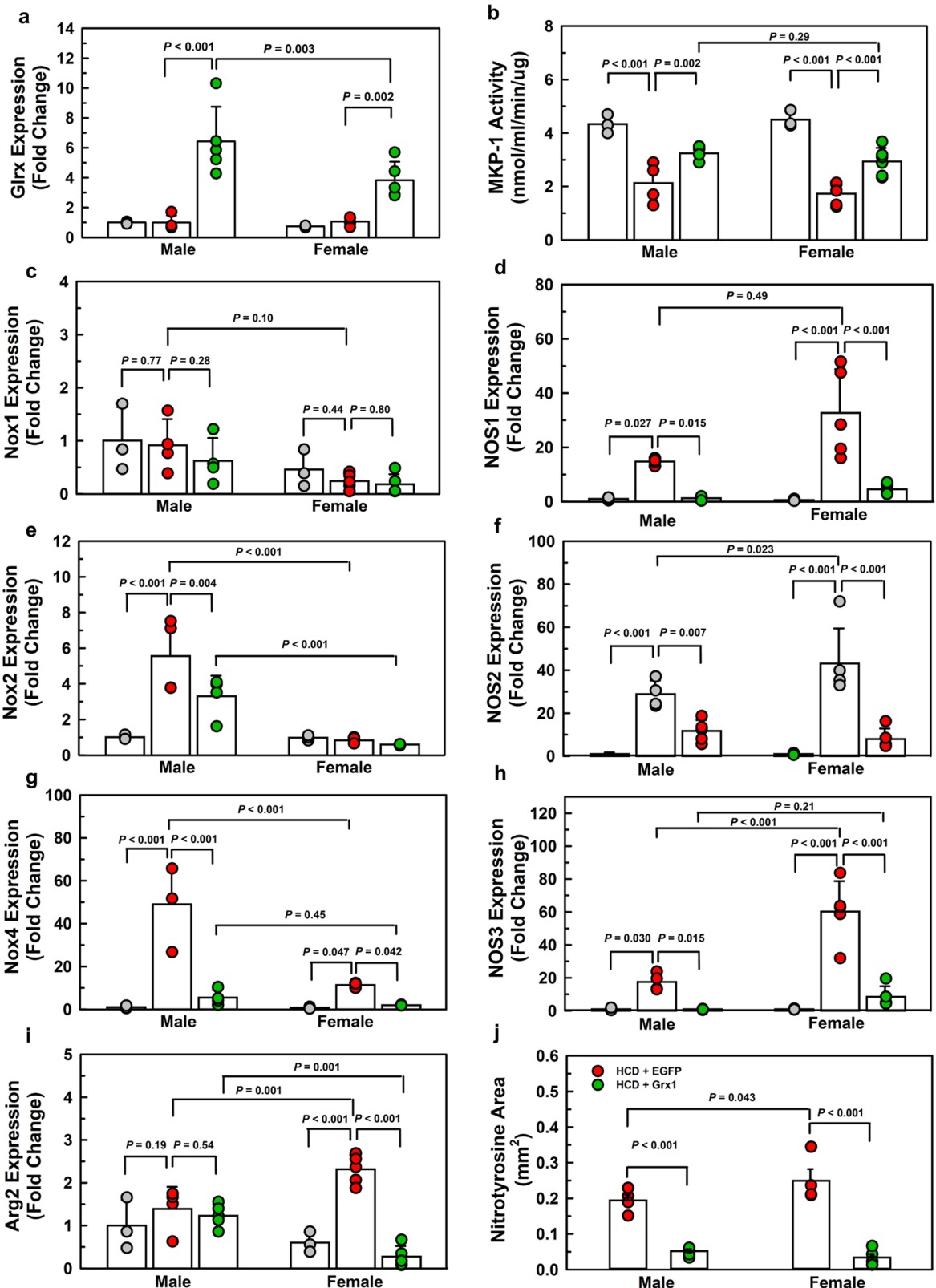

femur of male or female 10-week-old C57BL6/J mice, and hematopoietic progenitor cells were isolated by negative selection using a Robosep automated cell separator (Stem Cell Technologies) and the Mouse Hematopoietic Progenitor Cell Isolation Kit (Stem Cell Technologies). Isolated cells were resuspended in RPMI supplemented with 1% Penicillin/Streptomycin, stem cell factor (100 ng/ml), Flt-3L (100 ng/ml), IL-11 (100 ng/ml), and IL-3 (20 ng/ml), and transduced with lentivirus (MOI = 30) by incubating overnight at 37 °C in a $CO_2$ incubator. The transduced hematopoietic progenitor cells were washed with PBS and resuspended

at a concentration of 5 ×10[5] cells/100 μl. BM recipient mice were randomized into two groups (EGFP$_{Mac}$$^{tg}$: $n = 9$; Grx1$_{Mac}$$^{tg}$: $n = 10$) and were injected with virus-transduced cells of the corresponding sex via the retro-orbital sinus. Animals reconstituted with cells expressing CD68-EGFP were designated as EGFP$_{Mac}$$^{tg}$ and those that received CD68-Grx1-IRES-EGFP as Grx1$_{Mac}$$^{tg}$.

**Irradiation and bone marrow transplantation**. BM transplantations (BMT) were conducted as described previously[45]. Briefly, two weeks before irradiation and BMT,

**Fig. 10 Macrophage-restricted overexpression of Grx1 in LDLR[-/-] mice reverses the sexually dimorphic expression of NOX, NOS, and ARG2 genes induced by HCD feeding. a** Grx1 mRNA expression in bone marrow-derived macrophages isolated from 30-week-old male and female maintenance diet (MD)-fed LDLR[-/-] mice (**gray circles**, n = 3) and HCD-fed LDLR[-/-] mice overexpressing either EGFP (**red circles**) or Grx1 (**green circles**), n = 4–6 mice per group. **b** MKP-1 activity in bone marrow-derived macrophages isolated from 30-week-old male and female maintenance diet (MD)-LDLR[-/-] mice (**gray circles**, n = 3) and HCD-fed LDLR[-/-] mice overexpressing either EGFP (**red circles**) or Grx1 (**green circles**), n = 4–6 mice per group. **c–i** Gene expression profiling of NOX1, 2 and 4, ARG2 and NOS1, 2 and 3 in bone marrow-derived macrophages isolated from 30-week-old male and female maintenance diet (MD)-LDLR[-/-] mice (**gray circles**, n = 3) and HCD-fed LDLR[-/-] mice overexpressing either EGFP (**red circles**) or Grx1 (**green circles**), n = 4–6 mice per group. (**j**) Nitrotyrosine levels measured in the aortic root of 30-week-old male-female maintenance diet (MD)-LDLR[-/-] mice (**gray circles**, n = 3) and HCD-fed LDLR[-/-] mice overexpressing either EGFP (**red circles**) or Grx1(**green circles**), n = 4–6 mice per group, 8 aortic root sections per mouse. All data are expressed as mean ± S.D. One-way ANOVA followed by Fisher's Least Significance Difference test were used to compare the mean values between experimental groups. Source data are provided as a Source Data file.

male (n = 10) and female (n = 15) 10-week-old LDLR[-/-] mice were put on acidified water containing sulfamethoxazole (160 ng/ml) and trimethoprim (32 ng/ml). All designated transplant recipient mice were irradiated with 2 equal doses of 4.7 Gy, with 3 h between each dose (9.4 Gy total, Cobalt-60 Irradiator). Animals were allowed a 4 h recovery period prior to bone marrow transplantation. BM cell suspensions were collected from female C57BL/6 J and Grx1[−/−] donor mice by flushing the femurs and tibias with phosphate-buffered saline containing 2% fetal bovine serum (FBS, Invitrogen). Cells were pooled, washed, and resuspended in RPMI culture medium supplemented with 2 mmol/l L-alanyl-l-glutamine (GLU-TAMAX-1; Gibco BRL), 1% v/v nonessential amino acids (Gibco BRL), penicillin G/streptomycin (100 U/l and 100 μg/ml, respectively; Gibco BRL) and 2% FBS. Bone marrow cells from animals of the same strain (n = 3 per group) were pooled and washed twice with RPMI medium before centrifugation. Between washes, cells were passed through a cell strainer to clarify the samples. After washing, the cells were resuspended in RPMI culture medium (without FBS) and placed on ice until bone marrow injection. Irradiated LDLR[−/−] recipient mice were randomized into two groups based on the strain of their donors (n = 25/group) and BM cells (10–15 × 10[6] cells in 150–300 μl) were injected via the retro-orbital sinus. Animals reconstituted with C57BL/6 J BM are designated as wild-type (WT) and those that received Grx1[-/-] BM as Grx1$_{Leuko}$[-/-]. BM-recipients were fed a maintenance diet and allowed to recover for 4 weeks prior to initiating HCD feeding group. Both WT and Grx1$_{Leuko}$[-/-] mice were randomly subdivided into two groups receiving HCD for either 6 weeks (n = 10) or 20 weeks (n = 15). Due to dermatitis and weight loss, 2 mice from the 20-week HCD group (one WT mouse and one Grx1$_{Leuko}$[-/-] mouse) had to be prematurely euthanatized and excluded from the study. Complete blood counts were performed in all animals 6 or 20 weeks after BM transplantation at the time of euthanasia. For the gene transfer experiments, BM-recipients were fed a maintenance diet (MD) and allowed to recover for 4 weeks prior to initiating HCD feeding. Both WT and Grx1$_{Leuko}$[-/-] mice were randomly subdivided into two groups receiving HCD for either 6 weeks (n = 10) or 20 weeks (n = 15). To determine baseline gene expression and MKP-1 activity, we added a group of male and females LDLR[-/-] mice and fed them a maintenance diet for 24 weeks (n = 3).

**In vivo matrigel chemotaxis assay**. Each mouse received two Matrigel plugs three days prior to euthanasia as described in[7,8]. Briefly, subcutaneous injections of Matrigel (BD Biosciences) were made on the right and left flank of each mouse, one plug containing MCP-1 (500 ng/ml) and the other plug containing vehicle. After euthanasia, plugs were surgically removed, cleaned, and digested with dispase (BD Biosciences). Cells were stained with Calcein/AM (Invitrogen) and counted using an automated fluorescent cell counter (Nexcelcom Bioscience).

**Characterization of adipose tissue**. At the end of 18-month-old or 20 weeks on HCD, mice were anesthetized with isoflurane. Gonadal adipose fat pads were surgically removed through a ventral abdominal incision and weighted. 100 mg of each adipose tissue was homogenized with a glass pestle in 1 ml of TRIzol reagent and total RNA was isolated using PureLink RNA mini kit (Invitrogen). Reverse transcription was performed using SuperScript IV VILO (ThermoFisher). For F4/80 gene expression, RT-qPCR was performed using TaqMan@ gene expression assay (Mm00802529_m1) and the expression level was normalized to *Hprt* as the housekeeping gene. To evaluate the crown-like structure, adipose fat pads were embedded in paraffin, sectioned at a thickness of 5 μm and subjected to H&E staining.

**Analysis of atherosclerosis**. Two distinct vascular beds were used for the quantification of atherosclerosis, which was conducted as described previously[7,45,46]. After euthanasia, hearts and aortas were perfused with phosphate-buffered saline through the left ventricle. Hearts were separated from the aorta and embedded in Tissue-Tek Optimal Cutting Temperature compound (OCT) in a plastic cryomold (Tissue-Tek). OCT embedded hearts were rapidly frozen on dry ice and then stored at −80 °C until further processing. For *en face* analysis, aortas were dissected from the proximal ascending aorta to the bifurcation of the iliac artery and fixed with 4% paraformaldehyde in PBS. Adventitial fat was removed,

and aortas were opened longitudinally and stained with Oil Red O. Stained aortas were pinned flat onto black paper placed over dental wax, and digitally photographed at a fixed magnification. Total aortic area and lesion areas were calculated using Image-Pro Plus (Media Cybernetics). As a second measure of atherosclerosis, lesions in the aortic root from OCT embedded hearts were analyzed. Serial sections were cut through a 420-μm segment of the aortic root. For each mouse, 8 sections (10μm) separated by 80 μm were examined. Each section was stained with Oil Red O, and examined under a light microscope (Leica) with an attached digital camera. The aortic lesion area was quantified by averaging the total lesion area across sections using Image-Pro Plus (Media Cybernetics) and expressed as millimeters squared. To measure macrophage content in lesions, the slides containing aortic root sections adjacent to the Oil Red O-stained sections were dried overnight and fixed in ice-cold 100% methanol. Fixed aortic root sections were blocked with 3% BSA and stained with an anti-mouse CD68 antibody (Bio-Rad) overnight at 4 °C and the nuclear stain (DAPI, Invitrogen). Images were captured using a fluorescent microscope (Leica) and a high-resolution digital camera (Olympus). Macrophage content was calculated using Image-Pro software in each cryosection and expressed in millimeters squared. Non-specific staining was assessed using a rat IgG2a (Bio-Rad) antibody as the primary antibody.

**In situ labeling of *S*-glutathionylated proteins**. Aortic sections were fixed with 4% PFA and blocked free thiol groups with 40 mM N-ethylmaleimide in buffer containing 25 mM HEPESs pH 7.4, 0.1 mM EDTA pH 8.0, and 0.01 mM Neocuproine with 1% Triton (v/v). After three times washes with PBS, mixed disulfides were deglutathionylated with 27 μg/ml *E. coli* GRX1, 4 U/ml GSSG reductase (Roche), 1 mM GSH, 1 mM NADPH and 1 mM EDTA in Tris pH 8.0 for 20 min at RT. Aortic tissues were washed three times with PBS and newly reduced thiol were labeled with 1 mM *N*-(3-maleimidylpropionyl) biocytin (MBP, Sigma Aldrich) for 1 h at RT. Tissues were incubated with 10 ug/ml streptavidin-conjugated FITC (Thermo Fisher Scientific) for 1 h at RT and nuclei counterstain with DAPI (Invitrogen). The images were captured with a fluorescent microscope (Leica) and a high-resolution digital camera (Olympus).

**Nitrotyrosine immunohistochemistry**. For immunohistochemical analysis of nitrotyrosine production in aortic sections, anti-nitrotyrosine antibodies (Millipore, Cat. No. AB5411) were diluted 1:100 in 0.5% BSA in PBS and an IHC Select® HRP/DAB detection kit (Millipore) was used to stain the sections according to manufacturer's instructions.

**Plasma cholesterol, triglycerides, and lipoprotein profiles**. Mice were fasted overnight prior to euthanasia and blood was collected by cardiac puncture. Plasma cholesterol and triglycerides were quantified using enzymatic assay kits per the manufacturer's protocol (Wako Chemicals USA). For lipoprotein profiles, 110 ul plasma from each animal group was pooled, centrifuged to clarify the sample and 100 μl of pooled plasma was used for size exclusion chromatography. An ÄKTA FPLC and a Superose™ 6 10/300 column (GE Healthcare) were used at 20–25° C with a flow rate of 0.5 ml/min, as described previously[46]. Running buffer contained 1 mmol/L EDTA (EMD Millipore), 0.15 mol/l NaCl (Sigma-Aldrich®), and 0.02% wt/vol NaN3 (Sigma-Aldrich®), with the pH adjusted to 8.0. Fractions of 500 μl were collected and stored at −20 °C prior to analysis. Lipoprotein profiles were analyzed using the Cholesterol E kit according to the manufacturer's instructions (Fujifilm Wako Chemical Corporation).

**Peritoneal macrophage isolation and culture**. Resident peritoneal cells were harvested from mice by peritoneal lavage with 10 ml of ice-cold RPMI (1:1 mixture of Hyclone RPMI 1640 and glucose-free RPMI 1640 from Cellgro) supplemented with 2 mmol/l L-alanyl-l-glutamine (GLUTAMAX-1; Gibco BRL), 1% v/v nonessential amino acids (Gibco BRL), penicillin G/streptomycin (100 U/l and 100 μg/ml, respectively; Gibco BRL) and 2% FBS, and then purified by negative selection using antibody-coated magnetic beads (Dynabeads® mouse pan B (B220) and Dynabeads® mouse pan T (Thy 1.2)). Peritoneal cavities were lavaged twice, with a total of 20 ml, to maximize recovery. Cells were centrifuged and resuspended in RPMI complete

containing 2% FBS and total counts/viability was determined by Trypan blue extrusion using an automated cell counter (Nexcelcom Bioscience). Peritoneal cells were plated in 6-well plates at a concentration of $1 \times 10^6$ cells/ml in RPMI culture medium supplemented with 10% FBS and cultured in a humidified atmosphere at 5% CO$_2$. After 6 hours, non-adherent cells were removed through washing with RPMI culture medium leaving adhered peritoneal macrophages. All macrophages were maintained in culture for 24 hours in RPMI medium containing 10% FBS prior to any treatment.

**Metabolic priming of macrophages**. Metabolic priming was induced by incubating macrophages in RPMI supplemented with 10% FBS 100 µg/ml freshly isolated human LDL and 20 mmol/l D-glucose (LDL + HG) for 24 h.

**MKP-1 assay**. The MKP-1 activity assay was modified from the protocol established in our lab (Kim et al. 2016). Briefly, phosphatase release from a phosphotyrosine peptide (100 mM PTP) was measured from cell lysates (3 µg protein) in the presence or absence of MKP-1 inhibitor (40 mM sanguinarine). The amount of inorganic phosphate released was assayed spectrophotometrically using a Versa-Max spectrophotometric plate reader (Molecular Devices). Sanguinarine-sensitive phosphate released by MKP-1 was quantified with a standard curve prepared with known amounts of KH$_2$PO$_4$ (Malachite Green Assay, Cayman Chemical).

**Chemotaxis assay**. Purified peritoneal macrophages were cultured in Teflon bags under non-adherent conditions and metabolically primed for 24 h in RPMI culture medium (1:1 mixture of Hyclone RPMI 1640 and glucose-free RPMI 1640 from Cellgro) containing 10% FBS with human LDL + HG. Primed macrophages were loaded into the upper wells of a 48-well modified Boyden chamber (NeuroProbe). The lower wells contained either vehicle or 2 nM MCP-1 (R&D Systems). A 5 µm polyvinyl pyrrolidone-free polycarbonate filter membrane was layered between the upper and lower chambers, and the chamber was incubated for 3 h at 37 °C and 5% CO2. The membrane was washed and cells were removed from the upper side of the filter. Transmigrated cells were stained with propidium iodide (1 µM; Sigma) for 30 min at RT and cells were imaged and quantified using Kodak Image Station 4000MM (excitation 535 nm, emission: 600 nm) and Carestream Molecular Imaging software.

**Gene expression analysis**. Total RNA was isolated from purified peritoneal macrophages using PureLink RNA mini kit (Invitrogen). Isolated total RNA was quantified using a Nanodrop (Thermo Fisher), and 1 ug was treated with DNase I (Invitrogen) followed by being reverse-transcribed into cDNA using SuperScript IV VILO (Invitrogen), according to manufacturer's instructions. Quantitative real-time PCR experiments were performed using custom-designed TaqMan® Array Cards, 384-format (Thermo Fisher). Preamplified cDNA (100 ng) was mixed with TaqMan$^{TM}$ Fast Advanced Master Mix (Applied Biosystems). Thermal cycling was performed using the ViiA$^{TM}$7 real-time PCR system (Applied Biosystems). Relative gene expression was determined using the comparative $\Delta C_T$ method by comparing the $C_T$ values of a target gene for each sample with the indicated housekeeping genes (*Rn18S, Hprt, Tbp, Rpl13a, Ppia,* and *Gusb*).

Differential gene expression analysis was performed by principal component analysis (PCA) between male and female in WT and Grx1$^{-/-}$ mice using JMP Genomics 9. Analysis of relative gene expressions were used $\Delta\Delta C_T$ and Heatmaps were generated using web-based software, Morpheus (https://software.broadinstitute.org/morpheus).

**qPCR**. Total RNA was isolated from bone marrow-derived macrophages using PureLink RNA mini kit (Invitrogen). Isolated total RNA was quantified using a NanoDrop (Thermo Fisher). Reverse transcription was performed using SuperScript IV VILO (Invitrogen), according to the manufacturer's instructions. ST-QPCR was performed using TaqMan gene expression assay for *Glrx* (Mm00728386_s1), *Nox1* (Mm00549170_m1), *Cybb* (Mm01287743_m1), *Nox4* (Mm00479246_m1), *Arg2* (Mm00477592_m1), *Nos1* (Mm01208059_m1), *Nos2* (Mm00440502_m1) and *Nos3* (Mm00432403_m1). Expression levels were normalized to the housekeeping gene *Rn18S*.

**Statistical analyses**. Unless stated differently, data are expressed as mean ± standard deviation of the mean. Analysis of variance, followed by the Fisher's Least Significant Difference test, was used to compare the mean values between the experimental groups (SigmaPlot 14 software). $P < 0.05$ was set as the statistical significance level.

**Reporting summary**. Further information on research design is available in the Nature Research Reporting Summary linked to this article.

## Data availability

All relevant data are available from the corresponding author (R.A.). The source data underlying Figs. 1a–f, 2a, b, 3c, d, 4b, d, 5a, d, e, 6b, 7a–e, 8b, c, d–h, 9a–d, 10a–j, Supplementary Figures 1a–d, 2, 3a, b, 4a–c, 5a–d, 6c, 8a, b, 9a–d, 10 and Supplementary

Tables 1 and 3 are provided as a Source Data file with the manuscript. Source data are provided with this paper.

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

## Acknowledgements

We would like to thank Susan Foster, Jessica Han, Kevin Downs and Joe Cuellar for their technical assistance. This work was supported by grants the National Institutes of Health (HL115858 and HL153120) to R.A.

## Author contributions

R.A. conceived the study and designed the experiments. Animal studies, including gene transfer and bone marrow transplantation experiments and Matrigel plug assays, were conducted by Y.J.A. and S.T. MKP-1 activity assays were conducted by Y.J.A., H. N.N. and L.W. Histology and atherosclerotic lesion assessments were performed by Y.J.A. and S.T. Gene profiling was conducted by L.W., Y.J.A. and J.D.S. R.A. and Y.J.A. drafted the article and provided data analysis and interpretation. All authors provided critical revisions of the article and final approval of the version to be published. Authors have no financial or non-financial interest or conflicts to disclose.

## Competing interests

The authors declare no competing interests.
