## [Peer Review File · Nature Communications]

Reviewers' Comments:

Reviewer #1:

Remarks to the Author:

In this paper, the authors reported that the deletion of the thiol transferase glutaredoxin-1 (Grx) developed obesity, diabetes, and atherosclerosis only in aged female mice, not in male mice. Female mice showed higher infiltrations of macrophages in adipose tissues, suggesting Grx regulates macrophage priming only in females. Peritoneal macrophages from WT vs. Grx KO mice exhibited differential gene expression patterns between male and female, and female Grx KO macrophages highly expressed the oxidant source NOX4. The authors generated bone marrow-specific Grx KO mice in an LDLr KO background by bone marrow transplantation and demonstrated that myeloid-specific Grx deletion promoted obesity, hyperglycemia, and aortic lesions. As a likely mechanism, pro-inflammatory priming occurs in macrophages of female mice. Although the role of Grx in metabolism has been shown previously as Grx KO mice (different mouse background) developed fatty liver and hyperlipidemia, this study presents a novel study elucidating sex differences of Grx in macrophage biology. The study is exciting, but the cause of sexual dimorphism of Grx KO in aged male and female mice remains unclear.

Specific comments:

The authors should address the contribution of hormonal factors to the macrophage phenotype and regulation of Grx expression or activity. Is estrogen (should be low levels in post-menopausal females) or androgen modulate Grx/ glutathionylation levels in macrophages?

There is an experimental limitation in the BM transfer study. They only transferred BM from female Grx KO or WT mice to female LDLr KO mice. BM transfer from male Grx KO mice to female and male LDLr KO would confirm if epigenetic programming or hormonal control causes the sexual dimorphism of macrophages. They should clarify if male Grx KO macrophages do not promote atherosclerotic/metabolic phenotype in LDLr KO mice.

Figure 2: Is the data in panel A from aged WT and Grx KO mice? Matrigel plaques may also contain endothelial cells and other cell types. How did the authors confirm the cells are macrophages? Is the macrophage proportion/ differentiation the same in each group?

Figure 4e: For WT, there is only a "merged" photo, which looks just like DAPI.

Figure 5. The heat map shows differential gene expression between male and female WT macrophages. Especially ROS generators and glucose metabolism-related genes exhibited expression differences, but there was no change in NOX4 expression between male and female WT mice. What were the other genes differentially regulated by sex and Grx? In the text, macrophages isolated from male Grx-deficient mice upregulated NOX1, NOX4, nNOS, and iNOS, whereas macrophages from females only upregulated NOX4 and eNOS. In the figure, except for NOX4, the data are not shown. Please provide the data as supplementary information. If the hyper-induction of NOX4 is responsible for the stressed phenotype of female macrophages, the authors should compare ROS/RNS generation between male and female Grx-deficient macrophages.

Supplemental Table 1: Is there male and female data?

Supplemental Table 3: There is a viewing problem with overlapping tables.

Reviewer #2:

Remarks to the Author:

In this manuscript, Dr. Asmis and colleagues used both aged and young female mice to show that GRX1 deficiency is sufficient to reprogram monocytes to obesogenic and pro-atherogenic macrophages. They further report that GRX1 deficiency induced the expression of ROS and RNS generating enzymes at the transcriptional level, which may partly contribute to the increased obesity and cardiovascular risk. This is an interesting, novel study; however, as explained below, there are some major concerns.

Major concerns:

1. The H&E staining of adipose tissues (Figs. 2c, 2d, Supplemental Figs. 5a, 5b) were not convincing enough to demonstrate increased adipose inflammation. The authors may need to stain for CD68 as in Fig. 4f.

2. Data for eNOS were not shown but only discussed in the main text in page 10, lines 243-245. Also, peroxynitrate levels were not measured but the authors have claimed to observe increased levels in the abstract and main text.

3. Fig. 5a shows that genes associated with ROS generators were upregulated in both sexes, more so in males according to the heatmap. Also, the example gene shown in Fig. 5d show that Nox4 RNA levels were higher in both males and females, except that the upregulation was more in females compared to males. Further, it is one of the common genes significantly upregulated in both sexes as shown in Fig. 5c. The authors have previously demonstrated that Nox4 overexpression under nutrient stress is sufficient for monocyte priming. Taken together, these data go against the authors current interpretation of results that this is a female-specific phenomenon. Moreover, it also infers that under nutrient stress, this Nox4 overexpression may be augmented, even comparable to females, thereby leading to monocyte priming. The authors may need to discuss this further and test it experimentally.

4. The authors have tested the role of GRX1-null monocytes in HCD-fed LDLR-null female mice but for reporting sexual dimorphism, the authors may need to repeat their experiments (Figs. 7 and 8) on male mice as well. I understand that the authors did not observe any phenotype in males at the basal level (chow-fed mice) but on a nutrient stress and atherosclerosis-prone environment, this might change. Especially, compared to data from Fig. 8, it would be interesting to see if ex vivo male macrophages from GRX1-null mice did not behave like their female counterparts.

5. The coefficient of variation in lesions in mice is typically about 50%, much higher than the authors observed in Fig. 7. The authors might comment on this, since otherwise it will raise suspicions in the community. Also, the number of mice used to assess lesion area in Fig 4 (3-5 per group) is too low despite the significant differences observed (see recent AHA recommendations on studies of atherosclerosis in mice).

6. Monocytes are often characterized as pro-inflammatory or anti-inflammatory by levels of the antigen Ly6C. Similarly, macrophages are often characterized as M1 or M2. Such analyses would help to relate the results of this study to the atherosclerosis field.

Reviewer #3:

Remarks to the Author:

Ahn and colleagues have investigated the role of glutaredoxin 1 (Grx1)-deficiency in male and female mice, including its effects on body weight gain, blood glucose and atherosclerosis. The main findings suggest that in chow-fed, reproductively senescent C57Bl/6 female mice, but not in age-matched male mice, whole-body Grx1-deficiency causes weight gain, elevated blood glucose and arterial lipid accumulation, and that hematopoietic Grx1-deficiency in *Ldlr*^{-/-} female mice fed a high-fat diet mimics the phenotype of chow-fed female Grx1-deficient mice but appear earlier. The Grx1-deficiency also altered macrophage gene expression and function consistent with the mouse phenotypes. The authors conclude that loss of monocytic Grx1 activity disrupts the immunometabolic balance and derepresses sexually dimorphic oxidative stress responses in monocytes and macrophages, which may contribute to increased obesity and cardiovascular risk observed in postmenopausal women.

Although interesting, the study does not provide sufficient mechanistic insight into the sexual dimorphism, there are methodological issues, and some of the results appear to be overinterpreted.

Specific comments:

1. The manuscript does not include any information on the mechanism of the sexual dimorphism. Would gonadectomy mimic the effect in young female mice? Do female sex hormones act on this pathway in macrophages? If so, what is the mechanism? Would inhibition of androgen signaling in male mice reveal an effect on Grx1-deficiency?

2. The insulin resistance is not sufficiently well investigated. At the very least, insulin- and glucose tolerance tests should be done in all groups of mice. Glucose clamps would also be helpful in determining the tissues contributing to insulin resistance. Furthermore, the glucose levels in Grx1-deficient mice cannot be classified as "hyperglycemia" (100-120 mg/dL) in mice.
3. It is stated that the increased body weight is due primarily to an increased gonadal adipose tissue weight. Other adipose tissue depots do not appear to have been assessed, so these results seem overinterpreted.
4. Was there hepatic steatosis in the whole-body and hematopoietic Grx1-deficient mice? An increased level of hepatic triglycerides could lead to increased VLDL production.
5. The lipoprotein profiles seem to be incorrect. C57Bl/6 mice do not have high levels of IDL/LDL and have a high peak of HDL. It looks to this reviewer like fractions ~25-35 in figure 1f could be HDL. The same is true for supplemental figure 2. Analysis of ApoB and ApoA-I in the fractions would strengthen the data. Likewise, the lipoprotein profile in figure 6e looks unusual for Ldlr^{-/-} mice. Why is there no HDL? The VLDL peaks in figure 1f and figure 6e do not line up.
6. Likewise, the atherogenesis data in the C57Bl/6 mice are not convincing. Are the abdominal patches of stained material really lesions of atherosclerosis? Have those been sectioned and examined for presence of macrophages? The picture in figure 4a does not look like a lesion. Furthermore, it would be informative to show a histology section adjacent to the CD68-stained section to ensure that the positive staining is localized to macrophage-like cells. All immunohistochemistry results should be quantified.
7. It is mentioned that eNOS was induced in macrophages from Grx1-deficient mice. Were are these data? Have this, as well as the Nox4 data (Fig. 5d), been confirmed by real-time PCR in separate samples?
8. It is interesting that the mice with hematopoietic Grx1-deficiency exhibit a body weight phenotype so much earlier than the whole-body knockouts. Is this due to the diet, the Ldlr-deficiency or to suppression of the body weight phenotype by another tissue in the whole-body knockouts? These issues also beg for inclusion of male mice in the bone marrow transplant experiments, and for fat-fed mice in the whole-body knockout experiments. It is possible that male mice would show a similar phenotype to females in these settings.
9. The atherosclerosis characterization in Ldlr^{-/-} mice should include assessments of lesion morphology.
10. It is concluded in the abstract and at the end of the discussion that the findings may explain part of the differences in cardiovascular risk between men and women and the elevated obesity and cardiovascular risk among postmenopausal women. This conclusion appears to be overstated. Women are usually protected from cardiovascular events, as compared with men, prior to menopause, and then the risks are more similar between men and women after menopause. The mouse phenotypes do not appear to mimic this pattern.

Minor comments:

11. It is concluded on page 12 that Grx1-deficiency exacerbates HCD-induced monocyte priming by increasing MKP-1 S-glutathionylation and inactivation. No data are provided to demonstrate causality.
12. Correct gene nomenclature should be used throughout the manuscript (e.g. eNos is not a gene name).
13. Are the female mice really reproductively senescent at 4-5 months of age?
14. Are the cells shown in figure 2d macrophages? Scale bars should be included in all tissue photos.

15. Western data should be quantified and subjected to statistical analysis.

Response to Reviewers' Comments

We would like to thank the reviewers for their help and constructive comments. Based on their comments we made three major changes and additions to the manuscript, which we believe further strengthened our conclusions and greatly improved our manuscript:

- 1) We added more mice to the aging study and now have a total of 5-6 18-month old mice in each of the four groups. The data from the new set of mice fully corroborated our initial findings.
- 2) As requested by all three reviewers, we repeated the bone marrow transplantation experiments in male mice. These new data support the conclusion that the sex difference in atherosclerosis susceptibility revealed by Grx1 deficiency is mediated by hematopoietic cells, most likely monocyte-derived macrophages, rather than by sex hormones.
- 3) Our gene expression profiling combined with the newly added detection of nitrotyrosine in the plaque of aged female but not male Grx1 KO mice suggest that the higher propensity for atherosclerosis in female mice induced by Grx1 deficiency appears to be mediated by the specific induction of Nox4 and eNOS in female macrophages, and the resulting formation of peroxynitrate.

Changes to the text of the manuscript are highlighted in red. Below please find our responses to each of the reviewers' comments:

Reviewer 1

"Specific Comments:

In this paper, the authors reported that the deletion of the thiol transferase glutaredoxin-1 (Grx) developed obesity, diabetes, and atherosclerosis only in aged female mice, not in male mice. Female mice showed higher infiltrations of macrophages in adipose tissues, suggesting Grx regulates macrophage priming only in females. Peritoneal macrophages from WT vs. Grx KO mice exhibited differential gene expression patterns between male and female, and female Grx KO macrophages highly expressed the oxidant source NOX4. The authors generated bone marrow-specific Grx KO mice in an LDLr KO background by bone marrow transplantation and demonstrated that myeloid-specific Grx deletion promoted obesity, hyperglycemia, and aortic lesions. As a likely mechanism, pro-inflammatory priming occurs in macrophages of female mice."

1. "Although the role of Grx in metabolism has been shown previously as Grx KO mice (different mouse background) developed fatty liver and hyperlipidemia, this study presents a novel study elucidating sex differences of Grx in macrophage biology. The study is exciting, but the cause of sexual dimorphism of Grx KO in aged male and female mice remains unclear." **We believe that the new data we added to the manuscript makes a strong case for a hematopoietic origin of this sex difference and points to differences in male and female macrophages. Our gene profiling data points to differences in the expression of ROS and RNS generators between male and female macrophages as the underlying mechanism and the propensity of female macrophages to generate peroxynitrate as evidenced by increase nitrotyrosine levels in atherosclerotic lesions of aged female but not in aged male Grx1 KO mice (new Fig. 6).**
2. "The authors should address the contribution of hormonal factors to the macrophage phenotype and regulation of Grx expression or activity. Is estrogen (should be low levels in post-menopausal females) or androgen modulate Grx/ glutathionylation levels in macrophages?" **While it is possible that hormonal factors may contribute to the different macrophage phenotypes, the fact that the increased propensity of aged female Grx1 KO mice was recapitulated in young mice that received Grx1 KO bone marrow suggests, as mentioned above, a hematopoietic origin for this sex difference. This conclusion is further supported by the fact that we used purified peritoneal macrophages for the gene profiling data, which were cultured for 2 days in the absence of sex hormones prior to their analysis, ruling out any direct effect of sex hormones. This, of course, does not rule out that sex hormones play a role in determining the epigenetic landscapes of male and female macrophages and their precursors.**

3. "There is an experimental limitation in the BM transfer study. They only transferred BM from female Grx KO or WT mice to female LDLr KO mice. BM transfer from male Grx KO mice to female and male LDLr KO would confirm if epigenetic programming or hormonal control causes the sexual dimorphism of macrophages. They should clarify if male Grx KO macrophages do not promote atherosclerotic/metabolic phenotype in LDLr KO mice." **This is an excellent point. We have now repeated the BMT experiments in male LDLR^{-/-} mice using male Grx1 KO BM and found no effect of Grx1-deficient bone marrow on HCD-induced atherogenesis in male mice. As the reviewer indicated, these data strongly suggest that differences in epigenetic programming (of macrophages) between male and female mice rather than hormonal control underlie the observed sexual dimorphism.**
4. "Figure 2: Is the data in panel A from aged WT and Grx1 KO mice?" **Yes, the data shown in Figure 2 is from aged mice. We adjusted the figure title accordingly to clarify this point.**
5. "Matrigel plaques may also contain endothelial cells and other cell types. How did the authors confirm the cells are macrophages? Is the macrophage proportion/ differentiation the same in each group?" **We used single-cell Western blot analysis to show that on day 3 after injection of the Matrigel, over 70% of cells recruited by MCP-1 in to the Matrigel plugs are monocytes and macrophages, and fewer than 5% of cells were CD45⁻ cells. We have published the details of this protocol in JoVE (see reference 8).**
6. "Figure 4e: For WT, there is only a "merged" photo, which looks just like DAPI." **We have now added the corresponding panels.**
7. "Figure 5. The heat map shows differential gene expression between male and female WT macrophages. Especially ROS generators and glucose metabolism-related genes exhibited expression differences, but there was no change in NOX4 expression between male and female WT mice. What were the other genes differentially regulated by sex and Grx? In the text, macrophages isolated from male Grx-deficient mice upregulated NOX1, NOX4, nNOS, and iNOS, whereas macrophages from females only upregulated NOX4 and eNOS. In the figure, except for NOX4, the data are not shown. Please provide the data as supplementary information." **The requested data for eNOS is now shown in Figure 5e and the data for nNOS, iNOS and Nox1 is shown in the new Supplementary Figure 5.**
8. "If the hyper-induction of NOX4 is responsible for the stressed phenotype of female macrophages, the authors should compare ROS/RNS generation between male and female Grx-deficient macrophages." **The comprehensive analysis of ROS and RNS generated by these macrophages was beyond the scope of this study. However, to test our hypothesis that macrophages from female, but not male Grx1 KO mice are prone to peroxynitrate production, we quantified nitrotyrosine levels in aortic root lesions (see new Fig. 6). As predicted nitrotyrosine was only detected in atherosclerotic lesions from aged female Grx1 KO mice but not in the aortic roots of aged male Grx1 KO mice nor in aged male of female WT mice (new Fig. 6).**
9. "Supplemental Table 1: Is there male and female data?" **Yes, in the revised Supplemental Table 1 we now indicate which data is from male mice and which from female mice. We apologize for that oversight.**
10. "Supplemental Table 3: There is a viewing problem with overlapping tables." **Our apologies, but we did not find any issues with the tables.**

Reviewer 2

"In this manuscript, Dr. Asmis and colleagues used both aged and young female mice to show that GRX1 deficiency is sufficient to reprogram monocytes to obesogenic and pro-atherogenic macrophages. They further report that GRX1 deficiency induced the expression of ROS and RNS generating enzymes at the transcriptional level, which may partly contribute to the increased obesity and cardiovascular risk. This is an interesting, novel study; however, as explained below, there are some major concerns.

Major Concerns:"

1. "The H&E staining of adipose tissue were not convincing enough to demonstrate adipose tissue inflammation. The authors may need to stain for CD68 as in Fig. 4f." **We agree. For that reason we quantified macrophage content of the adipose tissue by measuring changes in F4/80 mRNA expression levels (see Fig. 2b) instead of re-staining selected sections with CD68.**
2. "Data for eNOS were not shown but only discussed in the main text in page 10, lines 243-245." **Data for eNOS is now shown in Figure 5e.** "Also, peroxynitrate levels were not measured but the authors have claimed to observe increased levels in the abstract and main text." **Peroxyntirite production was now assessed indirectly by measuring the levels of nitrotyrosine in atherosclerotic lesions (Fig. 6). Only sections from female Grx1 KO mice showed any significant staining for nitrotyrosine.**
3. "Fig. 5a shows that genes associated with ROS generators were upregulated in both sexes, more so in males according to the heatmap. Also, the example gene shown in Fig. 5d show that Nox4 RNA levels were higher in both males and females, except that the upregulation was more in females compared to males. Further, it is one of the common genes significantly upregulated in both sexes as shown in Fig. 5c. The authors have previously demonstrated that Nox4 overexpression under nutrient stress is sufficient for monocyte priming. Taken together, these data go against the authors current interpretation of results that this is a female-specific phenomenon. Moreover, it also infers that under nutrient stress, this Nox4 overexpression may be augmented, even comparable to females, thereby leading to monocyte priming. The authors may need to discuss this further and test it experimentally." **We respectfully disagree that these data go against the authros current interpretation. The difference in Nox4 induction is dramatically different, more than 9-fold higher in macrophages from aged female Grx1 KO mice than aged male Grx1 KO mice. Of all the Noxes, Nox4 is the only one whose activity is primarily transcriptionally regulated rather than at the level of complex assembly. Furthermore, in contrast to other Noxes, Nox4 generates primarily H₂O₂ (J Biol Chem. 2011 Apr 15;286(15):13304-13), which oxidizes protein thiols, resulting in their S-glutathionylation. In the absence of Grx1 activity, these protein thiol-glutathione mixed disulfides can no longer be resolved. As the reviewer correctly points out, we showed previously that overexpression of Nox4 (genetically or nutrient stress-induced) alone is sufficient to promote monocyte priming, dysfunction and hyper-chemotactic activity (references 3 and 5). We also showed that primed monocytes give rise to reprogrammed, proinflammatory and proatherogenic macrophages. Thus, the large difference in Nox4 induction between male and female macrophages triggered by Grx1 deficiency, would predict an equally dramatic difference in monocyte priming, and explains the increased chemotactic activity of monocyte-derived macrophage in aged female Grx1 KO mice (Fig. 2). However, we do not claim that Nox4 overexpression alone explains the observed sex difference. In fact, other ROS generators Nox1 and iNOS and nNOS are also differentially expressed between male and female Grx1 KO macrophages and may contribute to the sexual dimorphisms in atherogenesis. But the dramatic 648-fold overexpression of eNOS in macrophages from female Grx1 KO mice suggested another important mechanism, the formation of peroxynitrite, by which female Grx1 KO macrophages may accelerate atherogenesis. Indeed, we found high levels of nitrotyrosine, a reaction product of peroxyitrite with proteins, in lesions of aged female Grx1 KO, but not in aged male Grx1 KO or in aged male or female WT mice (new Fig. 6).**

4. “The authors have tested the role of GRX1-null monocytes in HCD-fed LDLR-null female mice but for reporting sexual dimorphism, the authors may need to repeat their experiments (Figs. 7 and 8) on male mice as well. I understand that the authors did not observe any phenotype in males at the basal level (chow-fed mice) but on a nutrient stress and atherosclerosis-prone environment, this might change. Especially, compared to data from Fig. 8, it would be interesting to see if ex vivo male macrophages from GRX1-null mice did not behave like their female counterparts.” **We repeated the BMT study as requested and show that the transfer of BM from male Grx1 KO into male LDLR^{-/-} does not affect atherogenesis (Suppl. Fig. 8+9) , suggesting, as indicted by reviewer 1, that differences in epigenetic programming (of macrophages) between male and female mice rather than hormonal control underlie the observed sexual dimorphism.**

5. “The coefficient of variation in lesions in mice is typically about 50%, much higher than the authors observed in Fig. 7. The authors might comment on this, since otherwise it will raise suspicions in the community. Also, the number of mice used to assess lesion area in Fig 4 (3-5 per group) is too low despite the significant differences observed (see recent AHA recommendations on studies of atherosclerosis in mice).” **We would like to point out that the error bars shown are for the standard errors of the means, not standard deviations, with an n-number of 14. For example, the coefficients of variation for the *en face* analysis at 6 weeks were 21% (WT) and 33% (Grx1 KO) and 22% (WT) and 49% (Grx1 KO) at 20 weeks data. These values are well within the range we routinely observe in our laboratory. We would also like to point out that 1) our laboratory has many years of experience in analyzing atherosclerotic lesions, 2) large variations in the *en face* analysis can be avoided by diligently removing any adventitial fat, and 3) for the analysis of lesions in the aortic root, we routinely section at least 640 μm segments yielding 8 sections per mouse. This approach reduces the variance seen in studies that use only 3 – 5 sections per mouse. The number of mice in the aging study was increased to 5-6 mice per group.**

6. Monocytes are often characterized as pro-inflammatory or anti-inflammatory by levels of the antigen Ly6C. Similarly, macrophages are often characterized as M1 or M2. Such analyses would help to relate the results of this study to the atherosclerosis field. **We appreciate the suggestions, but unfortunately, these classifications are overly simplistic, and, in the case of macrophage, not appropriate (please refer to Immunity, 2014; 41(1): 14–20). The fact that Principal Component Analysis of our macrophage gene profiling data showed three different clusters (see Fig. 5b) is a case in point. We prefer to characterize macrophage phenotypes based on their functionalities or functional changes, e.g. proatherogenic, inflammatory, inflammation resolving, etc.**

Reviewer 3

“Ahn and colleagues have investigated the role of glutaredoxin 1 (Grx1)-deficiency in male and female mice, including its effects on body weight gain, blood glucose and atherosclerosis. The main findings suggest that in chow-fed, reproductively senescent C57Bl/6 female mice, but not in age-matched male mice, whole-body Grx1-deficiency causes weight gain, elevated blood glucose and arterial lipid accumulation, and that hematopoietic Grx1-deficiency in Ldlr^{-/-} female mice fed a high-fat diet mimics the phenotype of chow-fed female Grx1-deficient mice but appear earlier. The Grx1-deficiency also altered macrophage gene expression and function consistent with the mouse phenotypes. The authors conclude that loss of monocytic Grx1 activity disrupts the immunometabolic balance and derepresses sexually dimorphic oxidative stress responses in monocytes and macrophages, which may contribute to increased obesity and cardiovascular risk observed in postmenopausal women.

Although interesting, the study does not provide sufficient mechanistic insight into the sexual dimorphism, there are methodological issues, and some of the results appear to be overinterpreted.

Specific comments:"

1. "The manuscript does not include any information on the mechanism of the sexual dimorphism. Would gonadectomy mimic the effect in young female mice? Do female sex hormones act on this pathway in macrophages? If so, what is the mechanism? Would inhibition of androgen signaling in male mice reveal an effect on Grx1-deficiency?" **Our new BMT data in young male mice would predict gonadectomy would have little to no effect. In fact, our new data strongly suggest a hematopoietic, likely epigenetic origin of the sex differences we observed in Grx1 KO mice rather than a (direct) sex hormone-mediated mechanism.**
2. "The insulin resistance is not sufficiently well investigated. At the very least, insulin- and glucose tolerance tests should be done in all groups of mice. Glucose clamps would also be helpful in determining the tissues contributing to insulin resistance. Furthermore, the glucose levels in Grx1-deficient mice cannot be classified as "hyperglycemia" (100-120 mg/dL) in mice." **We agree with the reviewer and now refer to the changes in blood glucose in female Grx1 KO as "mild elevations in fasted blood glucose levels". We also deleted the word "hyperglycemia" from the title of the paper.**
3. "It is stated that the increased body weight is due primarily to an increased gonadal adipose tissue weight. Other adipose tissue depots do not appear to have been assessed, so these results seem overinterpreted." **Correct, we did not assess weight changes in other fat depots. However, the gonadal adipose tissue is the largest fat pad in mice. We have now re-phrased and clarified that statement (see p. 6, line 12)**
4. "Was there hepatic steatosis in the whole-body and hematopoietic Grx1-deficient mice? An increased level of hepatic triglycerides could lead to increased VLDL production." **We did not specifically analyze the livers for steatosis. However, we did see minor but statistically significant increases in liver weights in both aged female Grx1 KO as well as female HCD-fed mice that received female Grx1 KO BM, but not in their male counterparts. We also found significant elevations in VLDL and IDL/LDL cholesterol in aged female Grx1 KO mice, indicating the possible onset of hepatic steatosis. However, these changes in lipoprotein profiles were not recapitulated in young female LDLR^{-/-} mice that received female Grx1-KO BM, possibly because these subtle differences may have been masked by the combined effects of HCD-feeding and LDLR deficiency.**
5. "The lipoprotein profiles seem to be incorrect. C57Bl/6 mice do not have high levels of IDL/LDL and have a high peak of HDL. It looks to this reviewer like fractions ~25-35 in figure 1f could be HDL. The same is true for supplemental figure 2. Analysis of ApoB and ApoA-I in the fractions would strengthen the data. Likewise, the lipoprotein profile in figure 6e looks unusual for Ldlr^{-/-} mice. Why is there no HDL? The VLDL peaks in figure 1f and figure 6e do not line up." **The reviewer is correct, the labels had accidentally shifted. All figures were corrected accordingly.**
6. "Likewise, the atherogenesis data in the C57Bl/6 mice are not convincing. Are the abdominal patches of stained material really lesions of atherosclerosis? Have those been sectioned and examined for presence of macrophages? The picture in figure 4a does not look like a lesion. Furthermore, it would be informative to show a histology section adjacent to the CD68-stained section to ensure that the positive staining is localized to macrophage-like cells. All immunohistochemistry results should be quantified." **Yes. All adventitial fat was carefully removed prior to ORO staining. The lesions were confirmed at higher magnification. The ORO staining in Fig. 4a clearly identified lipid deposits in the vessel. While these lesions are very small, these lipid deposits clearly identify them as early lesions. As requested, we also added images of sections adjacent to those shown in Fig. 4c (CD68 – immunofluorescence images)**

7. "It is mentioned that eNOS was induced in macrophages from Grx1-deficient mice. Were are these data?" **The eNOS data is now shown in Fig. 5e, the Nox1, iNOS and nNOS data are now shown in Supplementary Fig. 5.** "Have this, as well as the Nox4 data (Fig. 5d), been confirmed by real-time PCR in separate samples?" **The data were all obtained by qRT-PCR in custom-designed TaqMan® Array Cards.**
8. "It is interesting that the mice with hematopoietic Grx1-deficiency exhibit a body weight phenotype so much earlier than the whole-body knockouts. Is this due to the diet, the Ldlr-deficiency or to suppression of the body weight phenotype by another tissue in the whole-body knockouts? These issues also beg for inclusion of male mice in the bone marrow transplant experiments, and for fat-fed mice in the whole-body knockout experiments. It is possible that male mice would show a similar phenotype to females in these settings." **The more rapid weight increase in the female mice with hematopoietic Grx1-deficiency as compared to the aged female Grx1 KO mice is very likely due to both, i.e. the high calorie-diet and the much more pronounced hypercholesterolemia due to the LDLR-deficiency (compare Fig. 1e with Fig. 7c) . We showed previously in non-human primates that increases in plasma cholesterol correlates with and is likely a major driver of monocyte priming and dysfunction (see reference 19).**
9. "The atherosclerosis characterization in Ldlr-/- mice should include assessments of lesion morphology." **We quantified lesion size based on ORO staining, a well-established and widely accepted technique to assess lesion size, and measured macrophage content as a measure of the extent of lesion inflammation. We don't believe quantifying aspects of lesion morphology would provide any significant new insights not already provided by our data and conclusions. The purpose of the BMT experiments was to examine the origin of the sex differences we observed in the aged Grx1 KO mice.**
10. "It is concluded in the abstract and at the end of the discussion that the findings may explain part of the differences in cardiovascular risk between men and women and the elevated obesity and cardiovascular risk among postmenopausal women. This conclusion appears to be overstated. Women are usually protected from cardiovascular events, as compared with men, prior to menopause, and then the risks are more similar between men and women after menopause. The mouse phenotypes do not appear to mimic this pattern." **We respectfully disagree. Our new data confirm that Grx1 deficiency unmask a sexual dimorphism with respect to atherogenesis in mice. This sexual dimorphism appears to be of hematopoietic origin and expresses itself once female mice become reproductively senescent.**

"Minor comments:

11. It is concluded on page 12 that Grx1-deficiency exacerbates HCD-induced monocyte priming by increasing MKP-1 S-glutathionylation and inactivation. No data are provided to demonstrate causality." **We demonstrated the causality in our previous publications and provided the corresponded references in that section (references 3 and 34).**
12. "Correct gene nomenclature should be used throughout the manuscript (e.g. eNos is not a gene name)." **We corrected the gene nomenclature throughout the manuscript.**
13. "Are the female mice really reproductively senescent at 4-5 months of age?" **According to the literature (see reference 28) female mice become reproductively senescent at approximately 8 months of age. We added this statement with the reference to page 10, 3rd paragraph).**
14. "Are the cells shown in figure 2d macrophages? Scale bars should be included in all tissue photos." **The H&E stain will stain all leukocytes in the crown-like structures and is not specific for**

macrophages. Macrophage content was determined independently based on F4/80 mRNA expression using RT-qPCR in these fat pads (Fig. 2b). Scale bars are now included.

- 15. "Western data should be quantified and subjected to statistical analysis." As noted in the figure legend to the figure, the quantification of the Western blot data shown in Supplementary Figure 7 is presented in Fig. 9.**

Reviewers' Comments:

Reviewer #1:

Remarks to the Author:

The authors reported that the deletion of glutaredoxin-1 (Grx) developed obesity, diabetes, and atherosclerosis only in aged female mice, not in age-matched male mice. Female Grx KO mice, not male mice, showed higher infiltrations of macrophages in adipose tissues, suggesting Grx regulates macrophage priming only in females. Peritoneal macrophages from WT vs. Grx KO mice fed normal chow exhibited differential gene expression patterns between male and female. In this revised manuscript, they show female Grx KO macrophages upregulate NOX4 and eNOS expression (not in male), and hematopoietic-Grx deficient aged female mice promote nitrotyrosine staining in the aortic root as a marker of peroxynitrite, but not in male mice. Also, hematopoietic-Grx deficiency in LDLr KO background with high calorie diet accelerates metabolic phenotype in young female mice, but not male mice. As a mechanism, combining their previous study, they show decreased MKP-1 activity (glutathionylated target), increased chemotaxis and apoptosis of Grx-deficient monocytes in female mice. This study presents novel data which hematopoietic cell Grx regulates monocytes/macrophage biology and may indicate a mechanism of sexual dimorphism in the development of obesity and atherosclerosis. However, some parts of the data are not clear and disconnected to the reviewer. More discussion on the data may improve the manuscript. The clinical implication is overinterpreted.

Specific comments:

1. The authors interpret increased NOX4 and eNOS in female macrophages induce peroxynitrite and oxidative stress. However, in the text, "In male mice, Grx1 deficiency upregulated Nox1, Nox4, nNOS and iNOS (Supplementary Fig. 5), whereas Grx1 deficiency in macrophages from female mice upregulated, Nox4 and eNOS (Fig. 5d+e)", which indicates male Grx KO macrophages may produce more oxidants. In particular, iNOS and nNOS can produce high levels of peroxynitrite. It should be discussed.
2. Were BMTs performed from Grx KO mice to only LDLR^{-/-} background mice? Please clarify the background of mice in Fig S4, S6, S9. In summary and introduction, LDLR^{-/-} background is not mentioned at all. Generally, Figure legends should explain more details. Ex. Figure 1, mice fed normal chow? At which age the plasma lipids were measured, etc. Fig S7, is this about female macrophages?
3. The author group previously published that the glutathionylated proteins in macrophages show sexual dimorphism (ref 6). Does the glutathionylation and inactivation MKP-1 happen in only female macrophages? Does this explain eNOS and NOX4 upregulation?
4. How the more apoptotic macrophages become more priming and increase the number in adipose tissue?
5. The implication to human post-menopausal women needs more clinical data which show, for example, the lower Grx or higher NOX4 levels in monocytes from aged women compared to aged men.

Minor

These should be removed or corrected.

Representative Western blot images for data shown in supplemental figure 6 (page 25)

Representative Western blot images for data shown in figure 8 (page 27)

Reviewer #2:

Remarks to the Author:

While a number of questions remain, the authors have done a good job addressing my points.

Reviewer #3:

Remarks to the Author:

The authors have performed several new experiments to address the reviewers' comments and questions, and the manuscript is improved. However, several important issues remain. Of these, the most critical are:

1. Lack of sufficient mechanistic insight: The authors argue that since bone marrow transplants were sufficient in phenocopying the sexual dimorphism, the mechanism is due to changes (priming) of the hematopoietic cells (likely through epigenetic changes). Although this is interesting and logical, it does not really address the mechanism behind these sexual dimorphic changes in hematopoietic cells. The manuscript would be strengthened by additional mechanistic insight into factors responsible for the priming.

2. Data interpretation and scientific rigor: Several issues need to be addressed. For example, the lipoprotein profiles still look incorrect. The profile in figure 1f shows a large IDL/LDL peak and no HDL. Since these are C57BL/6 mice, there should be mostly HDL and very little LDL. The profiles in figure 7 (from *Ldlr*^{-/-} mice) show no HDL and very little LDL. This also is not consistent with previous findings. Furthermore, the red material in the artery wall in figure 4a is probably lipid, but not a lesion. The CD68 stain does not seem specific. Negative controls are needed for all IHC experiments, including in figure 6. In addition, there is usually much more variability in the atherosclerosis phenotype than what is reported. (A quick plot of some of the raw data results in larger error bars than those shown in the manuscript, even as SEM.) Please show the data as individual data points rather than bar graphs (or bar graphs and individual data points).

Response to Reviewers' Comments

We would like to thank once again the reviewers for their constructive and helpful comments. We made all the requested edits to the manuscript and added the requested additional information. We also added new data on MKP-1 activity in macrophages from male LDLR KO that received bone marrow from male Grx1 KO (**Fig. S8 C**). We believe these new data provide further insights into the molecular mechanisms that underlie the sex differences between male and female Grx1 KO mice and strengthen our conclusions.

Changes to the text of the manuscript are highlighted in red. Below please find our responses to each of the reviewers' comments:

Reviewer #1 (Remarks to the Author):

The authors reported that the deletion of glutaredoxin-1 (Grx) developed obesity, diabetes, and atherosclerosis only in aged female mice, not in age-matched male mice. Female Grx KO mice, not male mice, showed higher infiltrations of macrophages in adipose tissues, suggesting Grx regulates macrophage priming only in females. Peritoneal macrophages from WT vs. Grx KO mice fed normal chow exhibited differential gene expression patterns between male and female. In this revised manuscript, they show female Grx KO macrophages upregulate NOX4 and eNOS expression (not in male), and hematopoietic-Grx deficient aged female mice promote nitrotyrosine staining in the aortic root as a marker of peroxynitrite, but not in male mice. Also, hematopoietic-Grx deficiency in LDLR KO background with high calorie diet accelerates metabolic phenotype in young female mice, but not male mice. As a mechanism, combining their previous study, they show decreased MKP-1 activity (glutathionylated target), increased chemotaxis and apoptosis of Grx-deficient monocytes in female mice. This study presents novel data which hematopoietic cell Grx regulates monocytes/macrophage biology and may indicate a mechanism of sexual dimorphism in the development of obesity and atherosclerosis. However, some parts of the data are not clear and disconnected to the reviewer. More discussion on the data may improve the manuscript. The clinical implication is overinterpreted.

Specific comments:

1. The authors interpret increased NOX4 and eNOS in female macrophages induce peroxynitrite and oxidative stress. However, in the text, "In male mice, Grx1 deficiency upregulated Nox1, Nox4, nNOS and iNOS (Supplementary Fig. 5), whereas Grx1 deficiency in macrophages from female mice upregulated, Nox4 and eNOS (Fig. 5d+e)", which indicates male Grx KO macrophages may produce more oxidants. In particular, iNOS and nNOS can produce high levels of peroxynitrite. It should be discussed.

The reviewer makes a good point. Please note that we found no evidence for enhanced peroxynitrite formation in aged male Grx1 KO mice (Fig. 6) despite the induction of Nox1, iNOS and nNOS (the changes in mRNA levels for Nox4 were not statistically significant). With the exception of Nox4, ROS formation by Noxes is not regulated at the transcriptional level. Thus, elevated mRNA levels do not necessarily lead to increased Nox activity and ROS formation. However, the increase in Nox4 expression we observed (Fig. 5d) is unlikely to account for the peroxynitrate formation we detected as Nox4 predominately generates

H₂O₂, not superoxide, which is required for peroxynitrate formation. It would appear that the major difference between aged male and female Grx1 KO mice is the massive induction of eNOS in aged female Grx1 KO mice, which greatly exceeded the induction of NOS enzymes in aged male Grx1 KO mice. Increased expression of eNOS could account for increased NO production and thus peroxynitrite formation in aged female Grx1 KO mice. While we clearly show that peroxynitrate is formed in lesions of aged female Grx1 KO mice, we can only speculate on the underlying mechanism and the source of superoxide required. Please note that while Nox2 was not induced in either aged male or female Grx1 KO mice, this enzyme is the major source of superoxide in macrophages and may provide the superoxide needed for peroxynitrate formation in aged female Grx1 KO mice. We added this information to the discussion on page 10, 2nd paragraph.

2. Were BMTs performed from Grx1 KO mice to only LDLR^{-/-} background mice? **Yes.** Please clarify the background of mice in Fig S4, S6, S9. In summary and introduction, LDLR KO background is not mentioned at all. **We updated legends to figures S4, S6 and S9 as well the summary and introduction accordingly.** Generally, Figure legends should explain more details. Ex. Figure 1, mice fed normal chow? **We added the diet information to the legend of figure 1.** At which age the plasma lipids were measured, etc. **We add the ages of the mice to all relevant figure legends.** Fig S7, is this about female macrophages? **Yes. These data were obtained in macrophages from female Grx1 KO mice as noted in Figure 9. We now added this information to the legend of figure S7.**

3. The author group previously published that the glutathionylated proteins in macrophages show sexual dimorphism (ref 6). Does the glutathionylation and inactivation of MKP-1 happen in only female macrophages? Does this explain eNOS and NOX4 upregulation?

That is an excellent question. Yes, in contrast to females, MKP-1 was not further inactivated in male LDLR KO mice that received bone marrow from male Grx1 KO (data now shown in Fig. S8c). This result suggests that in males, HCD-induced S-glutathionylation and degradation of MKP-1 in macrophages was not affected by Grx1 deficiency, possibly because Grx1 itself may be inactivated under conditions of HCD-induced nutrient stress. The reviewer is correct in that the differential expression of Nox4 between male and female macrophages may therefore account for this sex difference as increased Nox4 expression in female macrophages would lead to enhanced H₂O₂ flux and thus further inactivation of MKP-1. We reported previously that nutrient stress-mediated inactivation of MKP-1 in macrophages requires the induction of Nox4 (Kim et al. PNAS 2012, ref. 3), suggesting that the 8.4-fold higher induction of Nox4 in female macrophages in response to HCD feeding may account for the differences in MKP1 activity between male and female Grx1 KO macrophages. Since MKP-1 regulates numerous macrophage signaling pathways, cellular functions and even their activation states, this dual specificity phosphatase can be considered a master regulator of macrophage function (Kim et al. FRBM 2017, ref. 36). The differences in MKP-1 activity may at least in part explain the differences in macrophage “phenotype” between males and female Grx1 KO mice and thus their propensity to develop atherosclerotic lesions.

Our hypothesis that the vastly different induction levels of Nox4 between aged male and female Grx1 KO macrophages in response to HCD may account for the

(patho)physiological differences between aged male and female Grx1 KO mice is further supported by our previous reports that protein S-glutathionylation patterns in macrophages from HCD-fed LDLR KO mice differ greatly between male and female mice, with 42% of glutathionylated proteins being uniquely expressed in female macrophages (Ullevig ARS 2016, ref. 35). This information was added on page 10, 2nd paragraph.

4. How the more apoptotic macrophages become more priming and increase the number in adipose tissue?

We believe the increase in macrophages in the adipose tissue we observed is mainly due to the increased recruitment of monocyte-derived macrophages triggered by the enhanced sensitivity of “primed” blood monocytes to chemoattractants. Grx1 deficiency enhances monocyte priming and thus their chemotactic activity due to the loss of MKP-1 activity (Fig. 9a).

We do not claim that monocyte-derived macrophages in adipose tissue are more apoptotic. They are likely more sensitive to proapoptotic signals because MKP-1 controls MAPK pathways that trigger apoptosis. We merely demonstrate that Grx1 KO macrophages are also more sensitive to 7-ketocholesterol-induced apoptosis because their basal MKP-1 activity is reduced compared to WT macrophages (Fig. 9a), providing the (predicted) mechanistic link between Grx1 deficiency, MKP-1 activity and altered macrophage function and survival.

5. The implication to human post-menopausal women needs more clinical data which show, for example, the lower Grx or higher NOX4 levels in monocytes from aged women compared to aged men.

That is an excellent suggestion. Unfortunately, we were unable to find any human studies that evaluated and compared Grx1 and Nox4 levels in (aged) human monocytes from women and men.

Minor

These should be removed or corrected.

Representative Western blot images for data shown in supplemental figure 6 (page 25)
Representative Western blot images for data shown in figure 8 (page 27)

We apologize for the incorrect figure references. The figure legend for Fig. 9 now reads “Representative Western blot images are shown in supplemental figure 7”, the one for Fig. S7 reads “Representative Western blot images for data shown in figure 9c+d”.

.

Reviewer #2 (Remarks to the Author):

While a number of questions remain, the authors have done a good job addressing my points.

We thank reviewer 2 for the kind comments.

Reviewer #3 (Remarks to the Author):

The authors have performed several new experiments to address the reviewers' comments and questions, and the manuscript is improved. However, several important issues remain. Of these, the most critical are:

1. Lack of sufficient mechanistic insight: The authors argue that since bone marrow transplants were sufficient in phenocopying the sexual dimorphism, the mechanism is due to changes (priming) of the hematopoietic cells (likely through epigenetic changes). Although this is interesting and logical, it does not really address the mechanism behind these sexual dimorphic changes in hematopoietic cells. The manuscript would be strengthened by additional mechanistic insight into factors responsible for the priming.

The two BMT experiments not only point to differences in epigenetic programming in hematopoietic cells as the underlying cause of the sex differences we report here, but also further support our proposed molecular mechanisms: the differential expression of ROS and RNS generators in macrophages and the resulting differences in ROS/RNS produced by these cells, particularly H₂O₂ and peroxynitrate (please see expanded discussion on page 10, 2nd paragraph).

HCD promote the S-glutathionylation, inactivation and subsequent degradation of MKP-1 (Kim et al PNAS 2012, ref. 3). As noted above in our response to Reviewer 1, in contrast to females, MKP-1 was not further inactivated in male LDLR KO mice that received bone marrow from male Grx1 KO (now shown in Fig. S8 C). This finding strongly suggests that in male mice, HCD-induced S-glutathionylated, inactivation and degradation of MKP-1 in macrophages was not enhanced by Grx1 deficiency. We therefore hypothesize that the differential expression of Nox4 between male and female macrophages may account for this sex difference as increase Nox4 expression in female macrophages would lead to enhanced H₂O₂ flux and enhanced reduction in MKP-1 activity. We reported previously that nutrient stress-mediated inactivation of MKP-1 in macrophages requires the induction of Nox4 (Kim et al. PNAS 2012, ref. 3), suggesting that the 8.4-fold higher induction of Nox4 in female macrophages in response to HCD feeding may account for the differences in MKP-1 activity in male versus female Grx1 KO macrophages. Since MKP-1 regulates numerous macrophage signaling pathways, cellular functions and even their activation states, this dual specificity phosphatase can be considered a master regulator of macrophage function (Kim et al. FRBM 2017, ref. 36). The differences in MKP-1 activity may explain at least in part the differences in macrophage “phenotypes” between males and female Grx1 KO mice and thus their propensities to develop atherosclerotic lesions.

Our hypothesis that the vastly different induction levels of Nox4 in male versus female macrophages may account for the observed (patho)physiological differences between aged male and female Grx1 KO mice is further supported by our previous reports that protein S-glutathionylation patterns in macrophages from HCD-fed LDLR KO mice differ greatly between male and female mice, with 42% of S-glutathionylated proteins being detected only in female macrophages (Ullevig ARS 2016, ref. 35).

While the nature of the epigenetic programming underlying the differential expression of ROS and RNS generators between macrophages from male and female Grx1 KO mice is

not clear at this time, addressing the detailed epigenetic mechanisms would be beyond the scope of this paper.

2. Data interpretation and scientific rigor: Several issues need to be addressed. For example, the lipoprotein profiles still look incorrect. The profile in figure 1f shows a large IDL/LDL peak and no HDL. Since these are C57BL/6 mice, there should be mostly HDL and very little LDL.

The reviewer is correct. The labels were unfortunately not corrected during the last revision. We now corrected the lipoprotein profiles which now corresponds exactly to those we published previously (see Ahn et al. J Nutr Biochem. 2020)

3. The profiles in figure 7 (from Ldlr^{-/-} mice) show no HDL and very little LDL. This also is not consistent with previous findings.

We respectfully disagree. The main peak in plasma from HCD-fed LDLR KO mice is by far the VLDL peak. This profile is consistent with previously published profiles from HCD-fed LDLR KO mice (see Kim et al. Sci Report 2016, ref. 37, and Sberna A-L. et al ATVB 2011, Fig.1, PMID 21778422)

4. Furthermore, the red material in the artery wall in figure 4a is probably lipid, but not a lesion.

We respectfully disagree. O Red O stains neutral lipids and lipid droplets, which together with (CD68-positive) foam cells, by definition, make up the earliest detectable atherosclerotic lesions, also referred to as intimal xanthomas.

5. The CD68 stain does not seem specific.

CD68 is located primarily in lysosome of macrophages, which at times makes CD68 staining look diffuse. But the total lack of CD68 staining in WT (C57BL6/J) mice (Fig 4e) demonstrates the specificity of the staining and as such serves as an ideal negative control.

6. Negative controls are needed for all IHC experiments, including in figures 4 and 6.

We already provide negative controls; male WT (C57BL/6J) mice serve as a negative control, in figures 4 and 6.

7. In addition, there is usually much more variability in the atherosclerosis phenotype than what is reported. (A quick plot of some of the raw data results in larger error bars than those shown in the manuscript, even as SEM.) Please show the data as individual data points rather than bar graphs (or bar graphs and individual data points).

We have once again verified our calculations and found no errors. As we mentioned in our previous response (to Reviewer 2), the error bars shown are for the standard errors of the means, not standard deviations, with an n-number of 14. For example, the coefficients of variation for the *en face* analysis at 6 weeks were 21% (WT) and 33% (Grx1 KO) and 22% (WT) and 49% (Grx1 KO) at 20 weeks. These values are well within the range we routinely observe in our laboratory. We would also like to point out that 1) our laboratory has many years of experience in analyzing atherosclerotic lesions, 2) large variations in the *en face* analysis can be avoided by diligently removing any adventitial fat, and 3) for the analysis of lesions in the aortic root, we routinely section at least 640 μ m segments yielding 8 sections per mouse. This approach reduces the variance seen in studies that use only 3 – 5 sections per mouse.

We feel adding symbols to a histogram/bar graph with error bars reduces the clarity of the graph, particularly with $n=14$, and adds little additional information (see graph below; same data as shown in Fig. 4b of our manuscript). Furthermore, this format is widely used for the representation of atherosclerosis data in mice. We therefore prefer to leave the graphs in their current format.

Reviewers' Comments:

Reviewer #1:

Remarks to the Author:

The authors responded well to my questions and improved the manuscript with MKP-1 data of male macrophages. Fig S8C is missing in the legend, though (P.27).

Reviewer #3:

Remarks to the Author:

Although interesting, the revised manuscript has several remaining weaknesses:

1. The authors have not sufficiently addressed the concerns about scientific rigor.

a. For example, they argue that C57BL/6 mice are used as controls for CD68 immunoreactivity in figure 4e-f. This control is not appropriate because wildtype mice have no lesions. An adjacent section from the Grx^{-/-} mouse in figure 4f is needed as control (using a control primary antibody of the same concentration). Currently, it appears that all cells in this section are CD68-positive. Such controls are needed for all immunohistochemistry experiments, including in figure 4c. Furthermore, oil red O and CD68 should be shown in adjacent sections. It does not look like the sections in 4a and 4c are from the same animal/lesion site. Therefore, the presence of fatty streaks is not convincing. The oil red O staining appears to be in the media (although it is difficult to see if there are any intimal cells in 4a).

b. The lipoprotein profile in figure 7e does not look like the examples from the literature provided by the authors. It looks like most of the fractions were below reliable detection, so that only the VLDL peak is visible. LDLR^{-/-} mice fed a high fat diet should have a clear HDL peak and an IDL/LDL peak that appears as a shoulder on the VLDL peak (as in the example provided by the authors; figure 1 of PMID: 21778422).

c. Lesion data should be presented as scatter plots. This does not reduce the clarity, but rather increases transparency. Moreover, it is hard to understand how all data points can be within the SEM range in the example shown in the response to the reviewers. It might be useful to consult a statistician.

2. No additional experiments have been performed to increase the mechanistic insight into how sex differences result in different priming. For example, which sex hormones are responsible for the differences in Nox4 and eNOS? If not sex steroids, what other sex-specific mechanisms could be responsible for priming?

Response to Reviewer 3's Comments

We would like to thank once again Reviewer 3 for the constructive and helpful comments.

We increased the scientific rigor with additional histology controls, re-ran the FPLC lipoprotein profile and added new mouse experiments, overexpressing Grx1 in high-calorie diet fed mice to strengthen our conclusions. We believe the new data provide further insights into the molecular mechanisms that underlie the novel sex differences we reported between male and female mice with either whole-body, hematopoietic, or HCD-induced Grx1 deficiency.

We made all the requested edits to the manuscript and added the requested additional information. We also changed the order of Supplemental Figures 4 – 9 and updated the figure legends accordingly. Changes to the text of the manuscript are highlighted in red. Below please find our responses to each of the reviewers' comments:

Reviewer #3 (Remarks to the Author):

Although interesting, the revised manuscript has several remaining weaknesses:

1. The authors have not sufficiently addressed the concerns about scientific rigor.

a. For example, they argue that C57BL/6 mice are used as controls for CD68 immunoreactivity in figure 4e-f. This control is not appropriate because wildtype mice have no lesions. An adjacent section from the Grx^{-/-} mouse in figure 4f is needed as control (using a control primary antibody of the same concentration). Currently, it appears that all cells in this section are CD68-positive. Such controls are needed for all immunohistochemistry experiments, including in figure 4c. Furthermore, oil red O and CD68 should be shown in adjacent sections. It does not look like the sections in 4a and 4c are from the same animal/lesion site. Therefore, the presence of fatty streaks is not convincing. The oil red O staining appears to be in the media (although it is difficult to see if there are any intimal cells in 4a).

We repeated the staining experiments (Fig. 4) and added the requested controls, i.e. an adjacent section stained with a control rat IgG and the same secondary as in the section treated with antibodies directed against CD68. ORO and CD68 staining was done on adjacent sections.

We also chose a more representative section for the ORO staining (Fig. 4a) with ORO staining clearly present in the neointima (Fig. 4a, insert). The CD68-stained, adjacent section shows macrophage infiltration in the same areas (Fig. 4c).

*b. The lipoprotein profile in **figure 7e** does not look like the examples from the literature provided by the authors. It looks like most of the fractions were below reliable detection, so that only the VLDL peak is visible. LDLR^{-/-} mice fed a high fat diet should have a clear HDL peak and an IDL/LDL peak that appears as a shoulder on the VLDL peak (as in the example provided by the authors; figure 1 of PMID: 21778422).*

We re-ran the samples from these mice. The new FPLC profiles are shown in Figure 7e.

c. Lesion data should be presented as scatter plots. This does not reduce the clarity, but rather increases transparency. Moreover, it is hard to understand how all data points can be within the SEM range in the example shown in the response to the reviewers. It might be useful to consult a statistician.

As requested, we now present all lesion-derived data (oil red O staining; *en face* and in the aortic root as well as CD68 and nitrotyrosine staining) as scatter plots (Figs. 3c, 4b+d, 6b, 8b, d and h, 10j, Suppl. Figs. 7b and 10). We also reviewed the statistical analyses and corrected any errors we identified.

2. No additional experiments have been performed to increase the mechanistic insight into how sex differences result in

different priming. For example, which sex hormones are responsible for the differences in Nox4 and eNOS? If not sex steroids, what other sex-specific mechanisms could be responsible for priming?

We conduct additional experiments to further elucidate the mechanism(s) underlying the observed sex differences. Our data show that loss of Grx1 activity, either via genetic deletion or nutrient stress-mediated, promotes macrophage reprogramming and the induction, or derepression of sexually dimorphic expression patterns for NADPH oxidases, NO synthases and arginase 2, poisoning female macrophages for enhanced peroxynitrite formation and promoting atherogenesis, adipose tissue inflammation and weight gain. The sexual dimorphism is completely eliminated by overexpression of Grx1 in mice in a macrophage-restricted manner.

1. We added new data on the expression of arginase 2 in macrophages from aged male and female mice (Suppl. Fig. 5d). The increased expression of arginase 2 combined with robust induction of Nos3 seen only in macrophages from aged female Grx1^{-/-} mice strongly suggests NO synthase uncoupling in these cells and peroxynitrite formation. This hypothesis is supported by the nitrotyrosine staining observed only in the aortic roots of aged female Grx1^{-/-} mice but not in aged male or young Grx1 KO mice, and may explain the onset of lesion formation in aged female Grx1 KO mice. We observed a similar phenomenon in atherosclerosis-prone LDLR-KO mice, which in response to a high-calorie diet (HCD) showed stark sex differences in the induction of NADPH oxidases, NO synthases and arginase 2, and significantly more pronounced peroxynitrite-mediated protein damage in the aortic lesions of female mice than males. Importantly, the induction of ROS and RNS-generating enzymes in macrophages triggered by HCD feeding was completely reversed by overexpression of Grx1 in these cells, confirming a direct link between Grx1 activity and the sexually dimorphic expression of ROS and RNS-generating enzymes. Our data show that Grx1 deficiency is both necessary and sufficient to unmask the reported sex differences.
2. Our data show that Grx1 deficiency in the context of aging or induced by HCD-feeding results in the induction, or derepression, of ROS and RNS-generating enzymes. Our BMT experiments showed that deficiency of Grx1 in monocytes and macrophages recapitulates the effects of whole-body Grx1 deficiency and the sex difference observed in aging mice, strongly suggesting that epigenetic programming rather than the direct effects of sex hormones account for these sex differences. Our newest data demonstrate that overexpression of Grx1 in monocytes and macrophages is sufficient to reverse the sex differences unmasked by HCD feeding, i.e. the differential expression of ROS and RNS-generating enzymes. Reversing the sex difference in the expression of these genes protected against atherosclerosis by reducing oxidative damage within the vessel walls. Together our data identified a previously unknown sexual dimorphism and revealed a novel, critical role for monocytic Grx1 in protecting against atherosclerosis

Reviewers' Comments:

Reviewer #3:

Remarks to the Author:

The manuscript is much improved, but revisions are still needed.

1. The scatter plots should include indications of mean or median and error bars. Also, the symbols need to be smaller so that all individual data points are clearly visible. For example, in figure 3C only 1-2 data points are visible in the WT mice. Figure 3D still has not been converted to a scatter plot.

Response to Reviewer 3's Comments

We would like to thank once again Reviewer 3 for the constructive and helpful comments.

We made all the requested edits to the figures. We also added a graphical abstract to the Supplemental material as Supplemental Figure 11 and a legend to the Figure Legends. Below please find our responses to each of the reviewers' comments:

Reviewer #3 (Remarks to the Author):

The manuscript is much improved, but revisions are still needed.

*1. The scatter plots should include indications of mean or median and error bars. **Mean and error bars have now been included in all these figures***

*Also, the symbols need to be smaller so that all individual data points are clearly visible. For example, in figure 3C only 1-2 data points are visible in the WT mice. **Symbol size was reduced in all these figures.***

*Figure 3D still has not been converted to a scatter plot. **Figure 3D as well as all remaining bar graphs have now been converted***